# Complete mitochondrial genomes of four species of praying mantises (Dictyoptera, Mantidae) with ribosomal second structure, evolutionary and phylogenetic analyses

**Yan Shi**[1,2], **Lin-Yu Li**[1], **Qin-Peng Liu**[1], **Muhammad Yasir Ali**[1], **Zhong-Lin Yuan**[1], **Guy Smagghe**[2,3]*, **Tong-Xian Liu**[1]*

**1** Key Lab of Integrated Crop Pest Management of Shandong Province, College of Plant Health and Medicine, Qingdao Agricultural University, Qingdao, Shandong, China, **2** Department of Plants and Crops, Faculty of Bioscience Engineering, Ghent University, Ghent, Belgium, **3** Department of Biology, Free University of Brussels (VUB), Brussels, Belgium

* txliu@qau.edu.cn (TXL); guy.smagghe@ugent.be (GS)

## Abstract

Praying mantises are distributed all over the world. Though some Mantodea mitogenomes have been reported, an evolutionary genomic and phylogenetic analysis study lacks the latest taxonomic system. In the present study, four new mitogenomes were sequenced and annotated. *Deroplatys truncate*, *D. lobate*, *Amorphoscelis chinensis* and *Macromantis* sp. belong to Deroplatyidae, Amorphoscelidae and Photinaidae family, respectively. Our results indicated that the *ATP8* gene may be lost in *D. truncate* and *D. lobata* mt genome, and four tRNA genes have not been found in *D. truncate*, *D. lobata* and *Macromantis* sp. A dN/dS pair analysis was conducted and it was found that all genes have evolved under purifying selection. Furthermore, we tested the phylogenetic relationships between the eight families of the Mantodea, including 35 species of praying Mantis. Based on the complete mitochondrial genome data, it was also suggested as sister to Deroplatyidae + Mantidae, Metallyticus sp., the only representative of Metallyticidae, is sister to the remaining mantises. Our results support the taxonomic system of Schwarz and Roy and are consistent with previous studies.

## 1. Introduction

The praying mantis comprises up to 2,300 species having diversified morphological and ecological characteristics. These insects colonize at wider range of habitats, including arid and tropical rainforests, temperate regions and engaged in multiple hunting tactics [1, 2]. Praying mantises have significant applied importance in biological research such as agronomy, pharmacy and visualization. Many studies mainly focused on their biological properties, like the distribution, taxonomy, captive breeding behavior and the foodstuffs applications [3, 4].

The mitochondrial genome (mitogenome), as a robust molecular marker [5], has recently been utilized in the initial study of phylogenetic linkage among closely related species from Mantodea [6]. The typical insect mitogenome encodes a conserved set of 37 genes for 13

**Data Availability Statement:** Our data have been submitted to the NCBI database. In detail, The NCBI accession numbers are D. truncate (MT370514),

D. lobate (MT370513), A. chinensis (MT370512), and Macromantis sp. (MT370515).

**Funding:** We greatly thank Zeyi Lyu and Xianting Zhou provided the praying mantises specimens and ecology pictures. This study was supported by the National Nature Science Foundation of China Youth Fund (32001907). This study was also supported in part by Qingdao Agricultural University High-level Talent Fund (665-1117002; 663-1119002).

**Competing interests:** Co-Author Guy Smagghe is editor for the PLoS one journal and this does not alter the authors' adherence to all PLoS one Policies on sharing data and materials.

**Abbreviations:** *ATP6* and *ATP8* genes, ATPase subunits 6 and 8; COI, *COIII* genes for cytochrome coxidase subunits I-III; CYTB, apocytochrome b; *ND1-ND6* and *ND4L*, NADH dehydrogenase subunits 1–6 and 4L; *rrnL* and *rrnS*, large rRNA subunit and small rRNA subunit; PCGs, protein coding genes; mt, mitochondrial; BI, Bayesian inference; ML, maximum likelihood.

protein-coding genes (PCGs), two ribosomal RNA (rRNA) genes and 22 transfer RNA (tRNA) genes, having genome-level features, such as gene order, gene content and genome size, and some lineages exhibit great diversity [6, 7].

As per Schwarz's taxonomic system, Mantodea inheres 29 recognized families [8]. Amorphoscelidae comprises two subfamilies: Amorphoscelinae and Perlamantinae. *Amorphoscelis* belongs to the subfamily Amorphoscelinae, which can be distinguished from all other mantis through some characters such as: small, dorsoventrally flattened body, and adapted to a bark-living lifestyle [8]. Photinaidae comprises four subfamilies: Macromantinae, Photiomantinae, Cardiopterinae and Photinainae; *Macromantis* belongs to the Macromantinae subfamily [8]. Rivera & Svenson gave the Photinaidae morphological characters that they characterize [9]. Furthermore, Deroplatyidae is a lineage of primarily stick-like genera integrated due to genital morphology; widely oppressed [10, 11], the type genus of this family is *Deroplatys*. However, the phylogenetic relationships of Amorphoscelidae, Photinaidae and Deroplatyidae are also in doubt in the latest taxonomic system.

These four species belong to three families and inhabit quite different niches. Deroplatys' habitat environments are rainforest with high humidity. They perch in the thin branches and mimic dead leaves (S1A and S1D Fig). Amorphoscelis generally inhabits trunks, capturing small active insects such as ants or springtails (S1B Fig). Macromantis inhabits the dense broad leaves, and it requires high humidity and high temperature (S1C Fig). In this study, four new mitogenomes were described from different families (Amorphoscelidae, Photinaidae and Deroplatyidae), and a relative analysis of all available mitogenomes was conducted. These studies embodied a comprehensive analysis to elucidate the characteristics of protein-coding genes (PCGs), structural features in transfer RNAs (tRNAs) and ribosomal RNAs (rRNAs), the rate of evolution of PCGs, and the phylogenetic interrelation of these species based on 35 concatenated mitochondrial genes. Also, we discussed the possible reasons for *ATP8* gene loss in *D. truncate* and *D. lobata*. The results will lay the foundation for the study of Amorphoscelidae, Photinaidae and Deroplatyidae in the latest taxonomic system.

## 2. Results and discussion

The Illumina Hiseq 2500 platform with a PE150 strategy (150 base paired-end reads) was used to acquire four entire mitogenome sequences. A library having two indexes was constructed and sequenced by Genesky Biotechnologies (Shanghai, China). The four complete mitogenomes were assembled, and analyzed. Hence, the accuracy of the second-generation sequencing was confirmed by long-PCR.

### 2.1. Live habitus and mitogenome general features of newly sequenced Mantodea

We acquired 35 complete mantis mitogenomes in 8 families. Mt genomes of four species, *D. truncate* (MT370514), *D. lobate* (MT370513), *A. chinensis* (MT370512) and *Macromantis* sp. (MT370515), were sequenced and assembled from this study. Sampling and GenBank information are given in Table 1. The mt genomes of *D. truncate*, *D. lobate*, *A. chinensis* and *Macromantis* sp. have one typical annular chromosome (Fig 1). In *A. chinensis* and mt genome, 37 genes were identified that typically includes 13 protein-coding genes (*atp6*, *atp8*, *cob*, *cox1-cox3*, *nad4L*, *nad1-nad6*), two rRNA genes (*rrnL*, *rrnS*) and 22 tRNA genes (Fig 2 and S2 Table). However, in *D. truncate* and *D. lobate* mt genome, 32 of the 37 genes were identified including 12 protein-coding genes (*atp6*, *cob*, *cox1-cox3*, *nad4L*, *nad1-nad6*), two rRNA genes (*rrnS*, *rrnL*) and 18 tRNA genes (S8 and S9 Figs). In *Macromantis* sp. mt genome, 33 of the 37 genes were identified including 13 protein-coding genes (*atp6*, *atp8*, *cob*, *cox1-cox3*, *nad1-*

**Table 1. The information of samples used in this study.**

| Species | Family | Subfamily | Accession number | Reference |
|---|---|---|---|---|
| *Tropidomantis tenera* | Nanomantidae | Tropidomantinae | KY689127 | [26] |
| *Eomantis_yunnanensis* | Nanomantidae | Tropidomantinae | KY689138 | [26] |
| *Sceptuchus_simplex* | Nanomantidae | Nanomantinae | KY689133 | [26] |
| *Sphodromantis lineola* | Mantidae | Tenoderinae | KY689123 | [26] |
| *Tenodera_sinensis* | Mantidae | Tenoderinae | KY689132 | [26] |
| *Hierodulella_sp.* | Mantidae | Hierodulinae | KY689136 | [26] |
| *Hierodula_formosana* | Mantidae | Hierodulinae | KR703238 | [27] |
| *Rhombodera_valida* | Mantidae | Hierodulinae | KX611804 | [2] |
| *Tamolanica_tamolana* | Mantidae | Hierodulinae | DQ241797 | [28] |
| *Rhombomantis_brachynota* | Mantidae | Hierodulinae | KX611802 | [2] |
| *Hierodula_patellifera* | Mantidae | Hierodulinae | KX611803 | [2] |
| *Mekongomantis_quinquespinosa* | Mantidae | Hierodulinae | MN267041 | [29] |
| Mantidea_sp | Mantidae | / | KY689120 | [26] |
| *Mantis_religiosa* | Mantidae | Mantinae | KU201317 | [6] |
| *Statilia_sp.* | Mantidae | Mantinae | KU201316 | [6] |
| *Asiadodis_yunnanensis* | Mantidae | Choeradodinae | MN037794 | [30] |
| *Amantis_nawai* | Mantidae | Iridopteryginae | KY689114 | [26] |
| Humbertiella_nada | Gonypetidae | Gonypetinae | KU201315 | [6] |
| Theopompa_sp.-HN | Gonypetidae | Gonypetinae | KU201313 | [6] |
| *Deroplatys_lobata* | Deroplatyidae | Deroplatyinae | MT370513 | This study |
| *Deroplatys_truncata* | Deroplatyidae | Deroplatyinae | MT370514 | This study |
| *Deroplatys_desiccata* | Deroplatyidae | Deroplatyinae | KY689113 | [26] |
| *Metallyticus* sp. | Metallyticidae | / | KX434837 | [31] |
| *Hestiasula* sp. | Hymenopodidae | Oxypilinae | KY689115 | [26] |
| *Creobroter jiangxiensis* | Hymenopodidae | Hymenopodinae | KY689134 | [26] |
| *Odontomantis_sp.* | Hymenopodidae | Hymenopodinae | KY689121 | [26] |
| *Anaxarcha_zhengi* | Hymenopodidae | Hymenopodinae | KU201320 | [6] |
| *Creobroter_gemmatus* | Hymenopodidae | Hymenopodinae | KU201319 | [6] |
| *Theopropus_elegans* | Hymenopodidae | Hymenopodinae | KY689125 | [26] |
| *Creobroter_urbanus* | Hymenopodidae | Hymenopodinae | KY689137 | [26] |
| *Parablepharis_kuhlii_asiatica* | Hymenopodidae | Phyllothelyinae | KY689117 | [26] |
| *Phyllothelys_sp.2* | Hymenopodidae | Phyllotheliynae | KY689129 | [26] |
| *Phyllothelys_sp.1* | Hymenopodidae | Phyllotheliynae | KY689119 | [26] |
| *Amorphoscelis_chinensis* | Amorphoscelidae | Amorphoscelinae | MT370512 | This study |
| *Macromantis_sp* | Photinaidae | Macromantinae | MT370515 | This study |
| *Cryptocercus kyebangensis* | Blattoidea | Cryptocercidae | NC_030191 | [32] |

*nad6*, *nad4L*), two rRNA genes (*rrnS*, *rrnL*) and 18 tRNA genes (S10 Fig). Generally, bilaterian Mt genomes comprise two rRNA, 22 tRNA and 13 protein-coding genes (PCGs) on a single circular chromosome, with ~16kb in size [12, 13]. However, in many bilateral animals a diversion from the typical organization of the mt genome has obtained. For example, *atp8* gene is lost in most nematodes and thus only have 36 mitochondrial genes. In a species of tree frog (*Polypedates megacephalus*), the *nad5* gene is missing, and similarly, in many insects and other animals, some PCGs have also not been identified [7, 14–19]. In our study, the *atp8* gene is also absent in the *D. truncate* and *D. lobate* mt genome. There are several probabilities for lacking mitochondrial genes. First, it may be caused by the sequencing techniques and method, a mini-chromosome comprised the *atp8* gene, which is not identified. For example, the

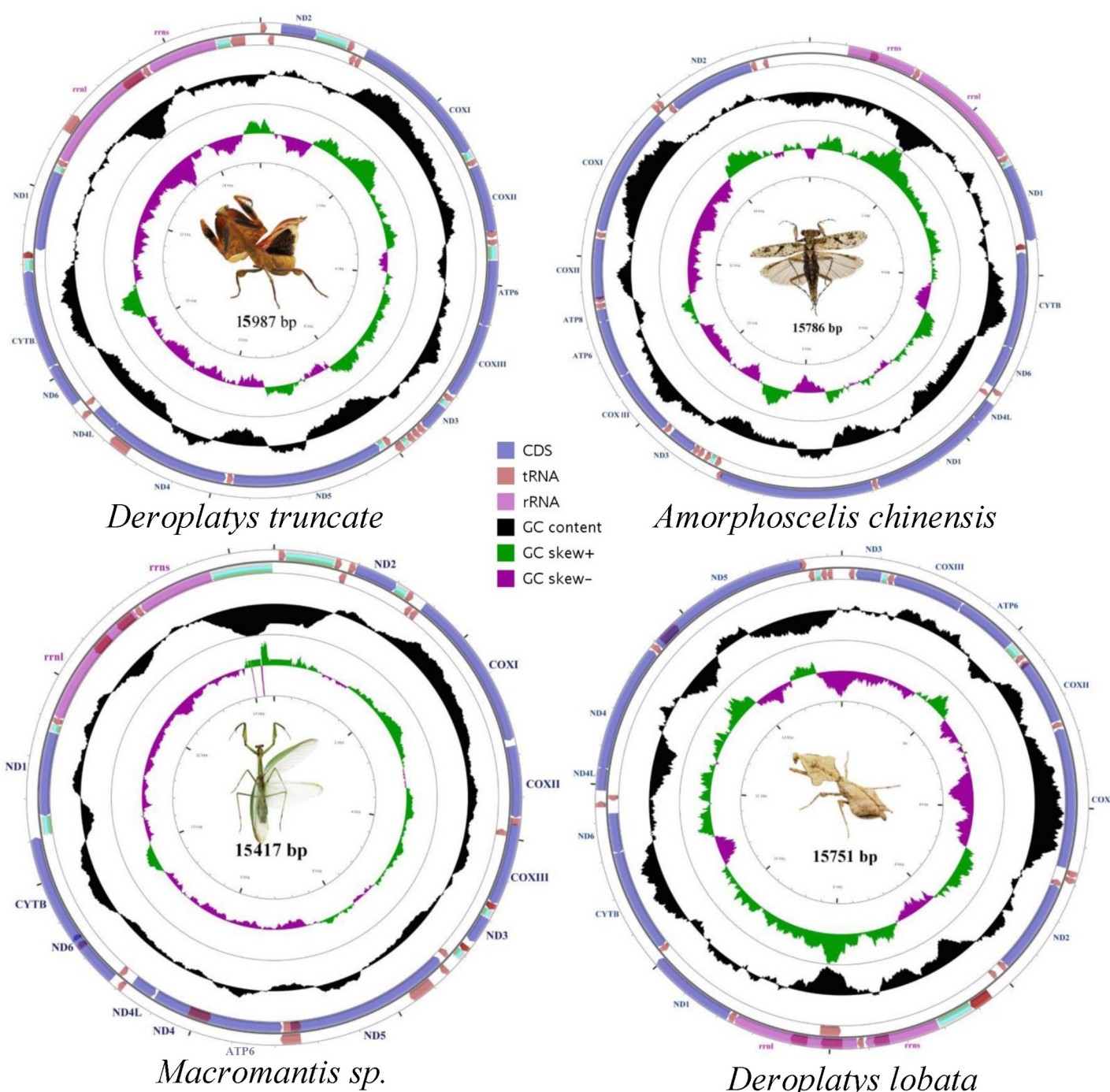

**Fig 1. Mitochondrial genomes of *Deroplatys truncate*, *Amorphoscelis chinensis*, *Macromantis* sp. and *Deroplatys lobata*.** Circular maps were drawn with CGView. Arrows indicate the orientation of gene transcription. Protein-coding genes are shown as blue arrows, rRNA genes as purple arrows, tRNA genes as brown arrows and non-coding regions as grey rectangle. Abbreviations of gene names are: *atp6* and *atp8* for ATP synthase subunits 6 and 8, *cox1-3* for cytochrome oxidase subunits 1–3, *cob* for cytochrome b, *nad1-6* for NADH dehydrogenase subunits 1–6, *rrnL* and *rrnS* for large and small rRNA subunits. The GC content is plotted using a black sliding window, as the deviation from the average GC content of the entire sequence. GC-skew is plotted as the deviation from the average GC-skew of the entire sequence. The inner cycle indicates the location of genes in the mitochondrial genome.

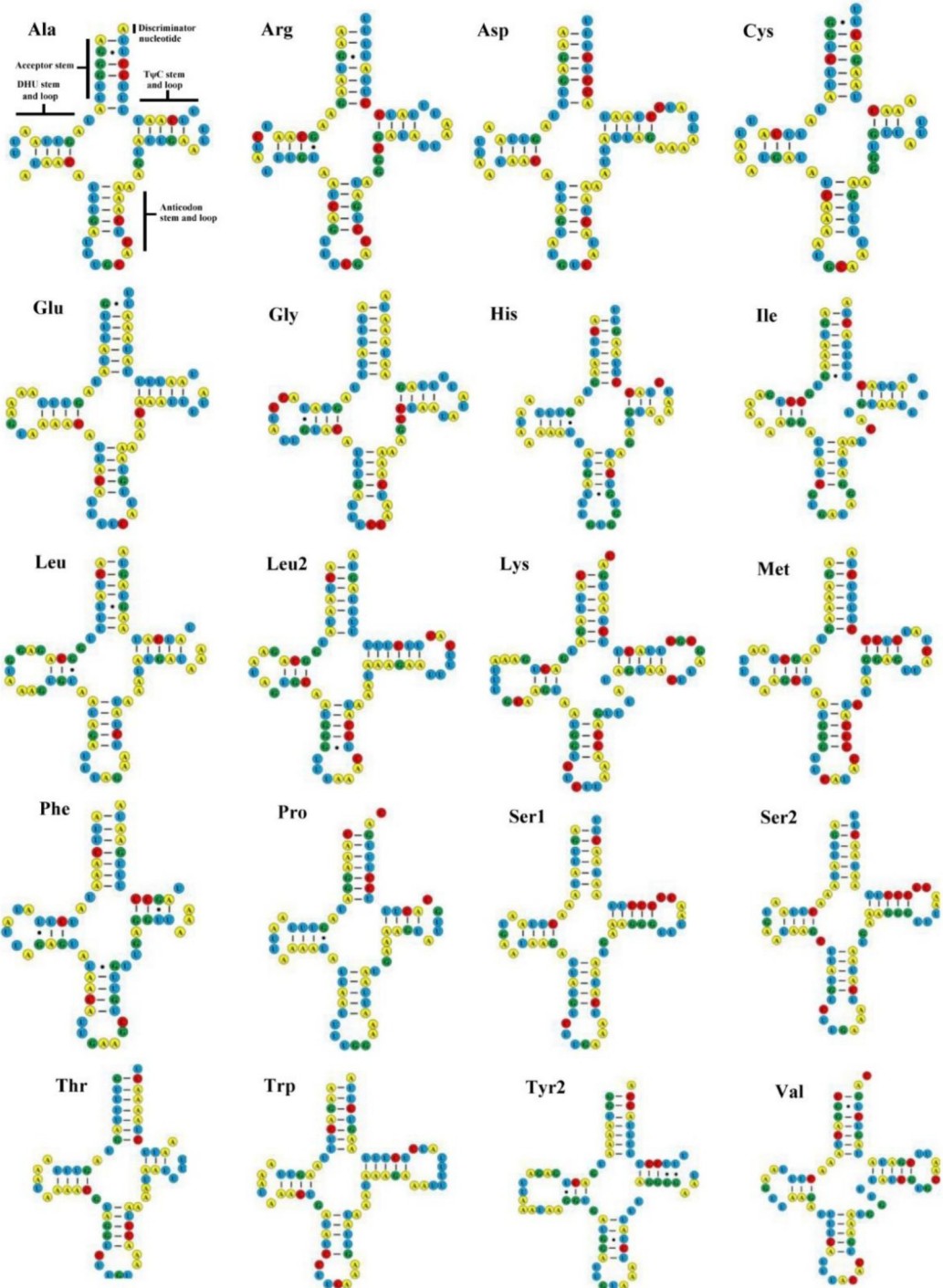

**Fig 2. Inferred secondary structures of 20 transfer RNAs (tRNAs) identified in *Amorphoscelis chinensis*.** Bars indicate Watson-Crick base pairings, and dots between G and U pairs mark canonical base pairings in RNA.

architecture of the mt genome in thrips (*Scirtothrips dorsalis*) exhibits an extreme asymmetry in chromosome size that only tRNA-Cys and *nad6* are on the 0.92 kb mini-annular chromosome [20]. Secondly, these lost genes are resulting exchanged by nuclear genes or generally relocate to the nuclear genome. Illustratively, mitochondrial *cox2* introgression has been

reported into the nuclear genome [21]. Thirdly, the lost genes have been gone in the long cycle of evolution. Since some protein-coding genes of mitochondrial genes are necessary in mito-chondrial respiration and the production of adenosine triphosphate, the deprivation of these genes would exhibit critical metabolism challenges to cells. For the undiscovered tRNA genes in the *D. truncate*, *D. lobate* and *Macromantis* sp., four tRNA genes were not found. The lack of tRNA genes additionally assorted. Illustratively, two species of gekkonids (*Uroplatus fimbriatus* and *U. ebenaui*), one species of *Caecilian amphibian* (Siphonopidae: Microcaecilia), an isopod crustacean (*Ligia oceanica*) and one booklouse species (*Liposcelis sculptilis*) are reported [7, 22–24]. Furthermore, among underneath metazoans, a huge loss of tRNA was described in cnidarians [25]. Overall, it is unclear whether in these cases the loss of genes is due to gene transfer to the nuclear genome or reflects the loss of the protein function.

## 2.2. Mitochondrial gene codon usage

Like other insects [19, 33], most PCGs use T or TAA as termination codons, whereas *D. truncate* and *D. lobate* COII and ND4L, *A. chinensis* ATP6 and ATP8, *Macromantis* sp. CYTB stops with TAG. The highest A+T content was present in all expanse, both non-coding and genes regions. Ala (A), Gly (G), Leu (L), Pro (P), Arg (R), Ser (S) Thr (T) and Val (V) are the most used and the frequency used for the codons of an amino acid was consistent in the four newly sequenced praying mantises (Fig 3). Furthermore, the difference in A+T content was also demonstrated further in the usage of codon (Table 2). Relative Synonymous Codon Usages (RSCU) presented that the four newly sequenced praying mantises used more NNT and NNA codon than NNC and NNG (Fig 3). These results are consistent with other published mantis mitogenomes [2, 19, 31, 34].

## 2.3. The secondary structure of ribosomal RNAs

The *rrnS* of *A. chinensis* included three structural domains (I-III) (Fig 4). The preserved sites were highlighted and analyzed within the 35 Mantodea species. The H47 was the most insecure among the eight helices of domain I (H9-H511). Domain II, comprising five helices (H567-H885), was extremely variable domain, specifically for helices H567, H577 and H673. In domain III, most of the helices were comparatively stable, except for H1068-H1113 and H1303. The *rrnL* of *A. chinensis* harbored five canonical structural domains (I-II, IV-VI) (Fig 5). The sites preserved in *rrnL* of the 35 Mantodea species were also analyzed. H563 was identically stable (74.6%). Domain II contained 14 helices (H579-H1196), and the degree of conservation was elevated in H671, H777, and H1087 (>76.0%). All helices belonging to Domains IV and V were comparatively conserved, apart from the helices H1648, H1764, H2077, H2259, H2395 and H2520 (<40.0%). Although the nucleotides including variable helices were greatly contrasting at the subfamily and family levels, and mainly molecules share certain alike secondary structures between species and are compensatory base changes (S2–S7 Figs). Our predicted secondary structure was consistent with the results of a previously published study [2].

## 2.4. Evolutionary rate and nucleotide diversity analysis

The non-synonymous/synonymous (dN/dS) exchange ratio can be utilized to evaluate whether a sequence is undergoing neutral, purifying or positive selection (dN/dS >1 is evidence for positive selection, <1 for purifying selection, and = 1 for neutral). For these eight mitogenomes, dN/dS pairwise analysis was performed. We found that all genes evolved under a free selection: cox1 revealed the most robust refined selection (0.028), while genes from the *nad* family (especially *nad6*) revealed a slightly relaxed purifying selection; *ATP8* was an outlier with a relatively low purification (0.31) (Fig 6). Nucleotide diversity analyses can be used to

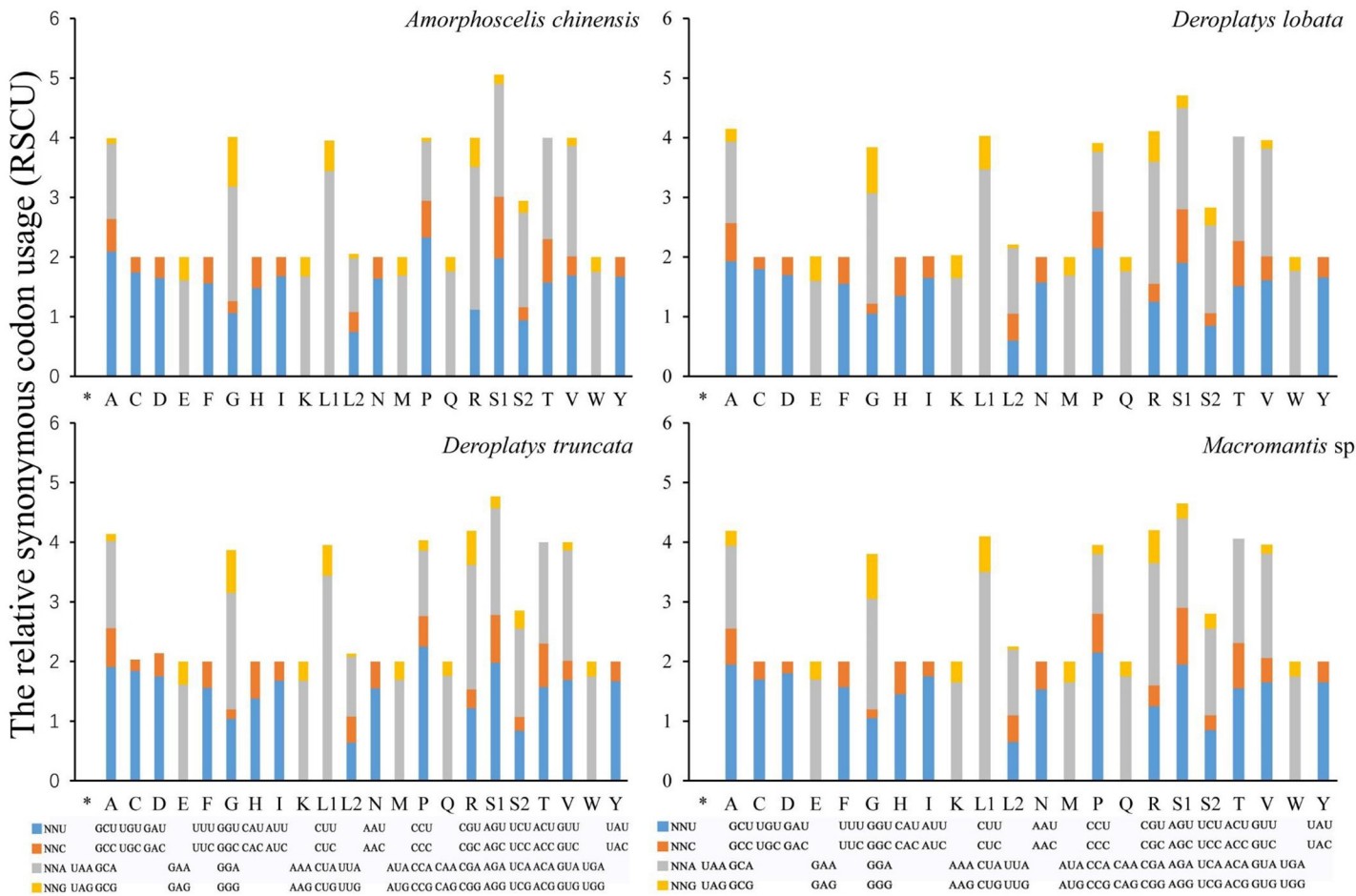

**Fig 3. Relative synonymous codon usage (RSCU) for protein coding genes of five booklice.** Abbreviations of tRNA genes are according to the single letter according to the IPUC-IUB one-letter amino acid codes.

design species-specific markers, particularly in taxa, where identification of morphological characters is complicated and ambiguous [6, 34], and it is also important for functional studies [6, 34, 35]. dN/dS pairwise analysis showed that cox1 is undergoing a robust purifying selection. The cox1 gene is constantly used as a universal barcode for species distinguishing in animals [36, 37], including insects [38–40]. Its low variability applied as barcode for the entire Mantidae, which must be intently tested and revised, assumes that its resolution power turns out to be too low. In that case, we suggest that genes exhibit an optimal amalgamation of rapid evolution and enough for large size, notably *nad6* and *nad4*, which should be estimated as potential DNA markers for the identification of population and species.

**Table 2.  A+T (%) composition of four mantodean mitochondrial genomes.**

| Species | Whole genome | A+T-rich region | PCGs |
|---|---|---|---|
| *D. truncate* | 75.8 | 82.3 | 76.0 |
| *D. lobate* | 75.6 | 80.7 | 77.9 |
| *A. chinensis* | 77.2 | 79.8 | 75.8 |
| *Macromantis sp.* | 76.5 | 81.2 | 73.0 |

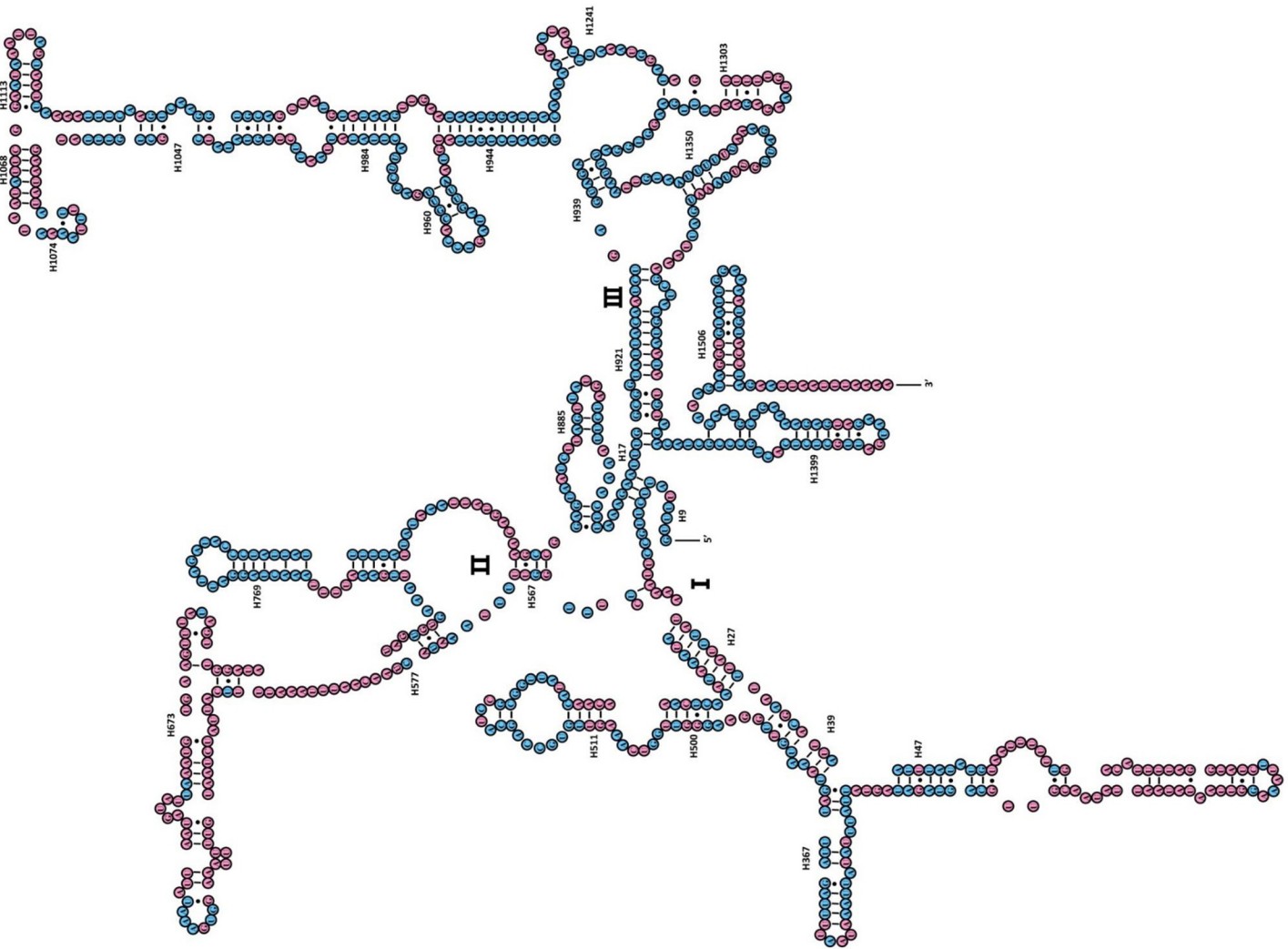

**Fig 4. Inferred secondary structure of rrnS of *Amorphoscelis chinensis*.** Inferred Watson-Crick bonds are illustrated by lines, whereas the noncanonical interactions are illustrated by dots.

## 2.5. Phylogenetic analyses on Mantodea

We evaluated the phylogenetic linkage among the prime lines of the Mantodea including 35 praying mantis species (Fig 7). Derived from the two contrasting datasets (a concatenated amino acid dataset and a concatenated nucleotide dataset), the phylogenetic link concluded from BI analyses shared the alike topologies (Fig 7). *Metallyticus* sp., the sole paradigmatic of Metallyticidae, is a sister to the other mantises, which are frequently studied as the primitive groups of the mantodean phylogeny [1, 10, 29–31, 41, 42]. For the determination of the level of the Deroplatyidae family, due to the head (without vertex process, highly developed juxta-ocular bulges), pronotal (tubercles, foliaceous expansion) and genital characteristic, Schwarz and Roy's studies strongly indicate that Deroplatyidae is in unstable phylogenetic position and represents a well-differentiated clade by multiple morphological data. Previous studies suggested that Deroplatyidae should correspond to a family rank [8, 10]. Based on the complete data of the mitochondrial genome, it was suggested that there should be a classification between Mantoidea as a sister to Deroplatyidae + Mantidae (Fig 7). In the latest classification system,

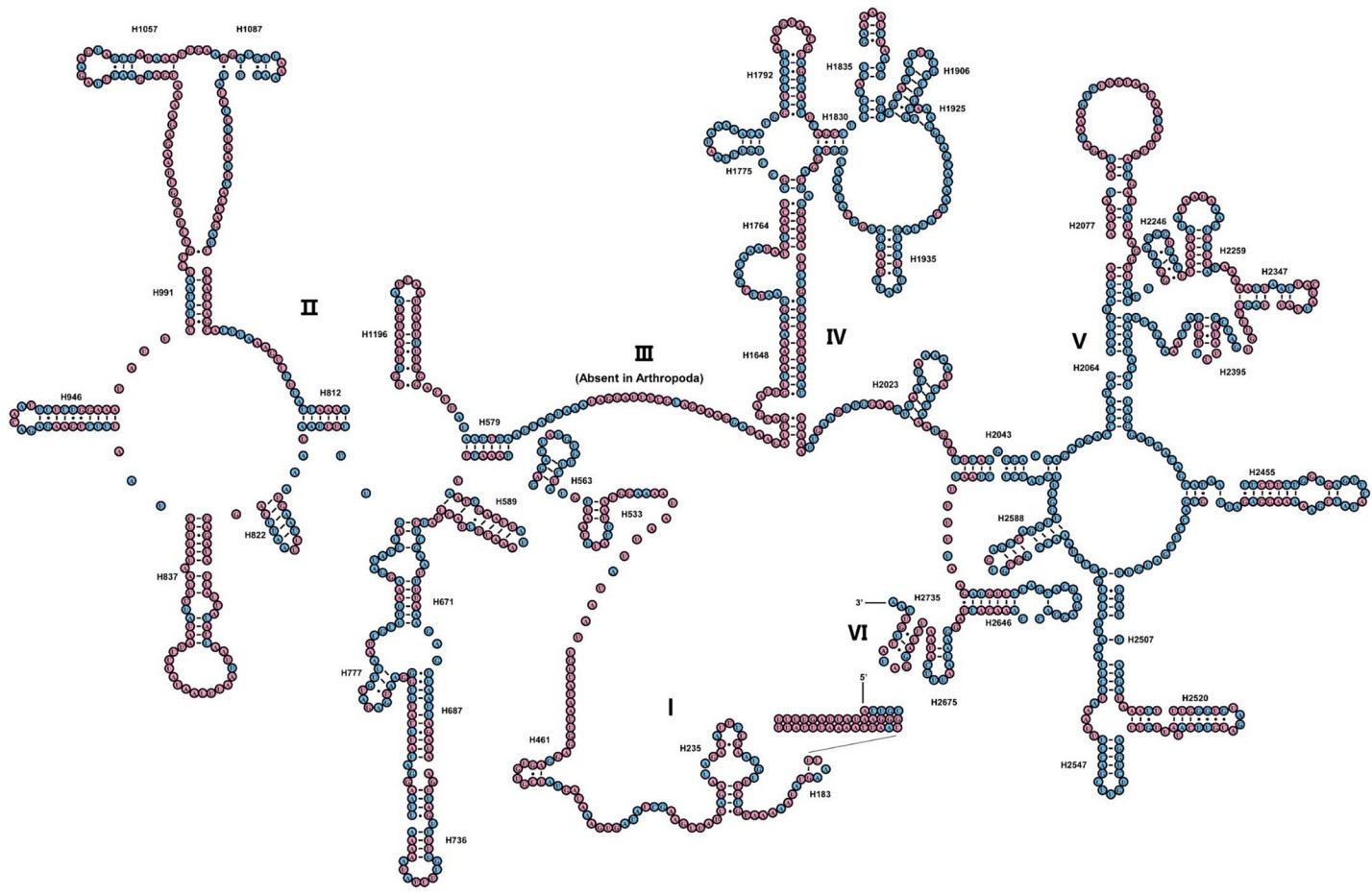

**Fig 5. Inferred secondary structure of rrnL of *Amorphoscelis chinensis*.** Inferred Watson-Crick bonds are illustrated by lines, whereas the noncanonical interactions are illustrated by dots.

*Macromantis* belongs to the subfamily Macromantinae, family Photinaidae, and *Amorphoscelis chinensis* belongs to the subfamily Amorphoscelinae, family Amorphoscelidae [8]. Interestingly, two species (Mantidae sp. + *Amantis nawai*) of Mantidae + Photinaidae (*Macromantis* sp.) are a sister group. In the future, more Photinaidae species are needed to confirm the phylogenetic status of this family rank. Furthermore, our results suggest that Amorphoscelidae and Nanomantidae are closely related. However, there are only one representative species of Amorphoscelidae, and molecular data from more species will be needed to determine the phylogenetic relationship in the future.

## 3. Materials and methods

### 3.1. Taxon sampling

The latest Mantodea taxonomic system, Schwarz's system [8], was used. Thirty-one complete mt genomes represented five families (Hymenopodidae, Mantidae, Gonypetidae, Nanomantidae, and Metallyticidae) on GenBank before this study. The complete mitogenomes of *D. truncate*, *D. lobate*, *A. chinensis*, and *Macromantis* sp., belonging to Deroplatyidae, Amorphoscelidae, and Photinaidae, respectively, were amplified and sequenced. Thirty-five mantis species, including 8 families were sampled: Hymenopodidae, Photinaidae,

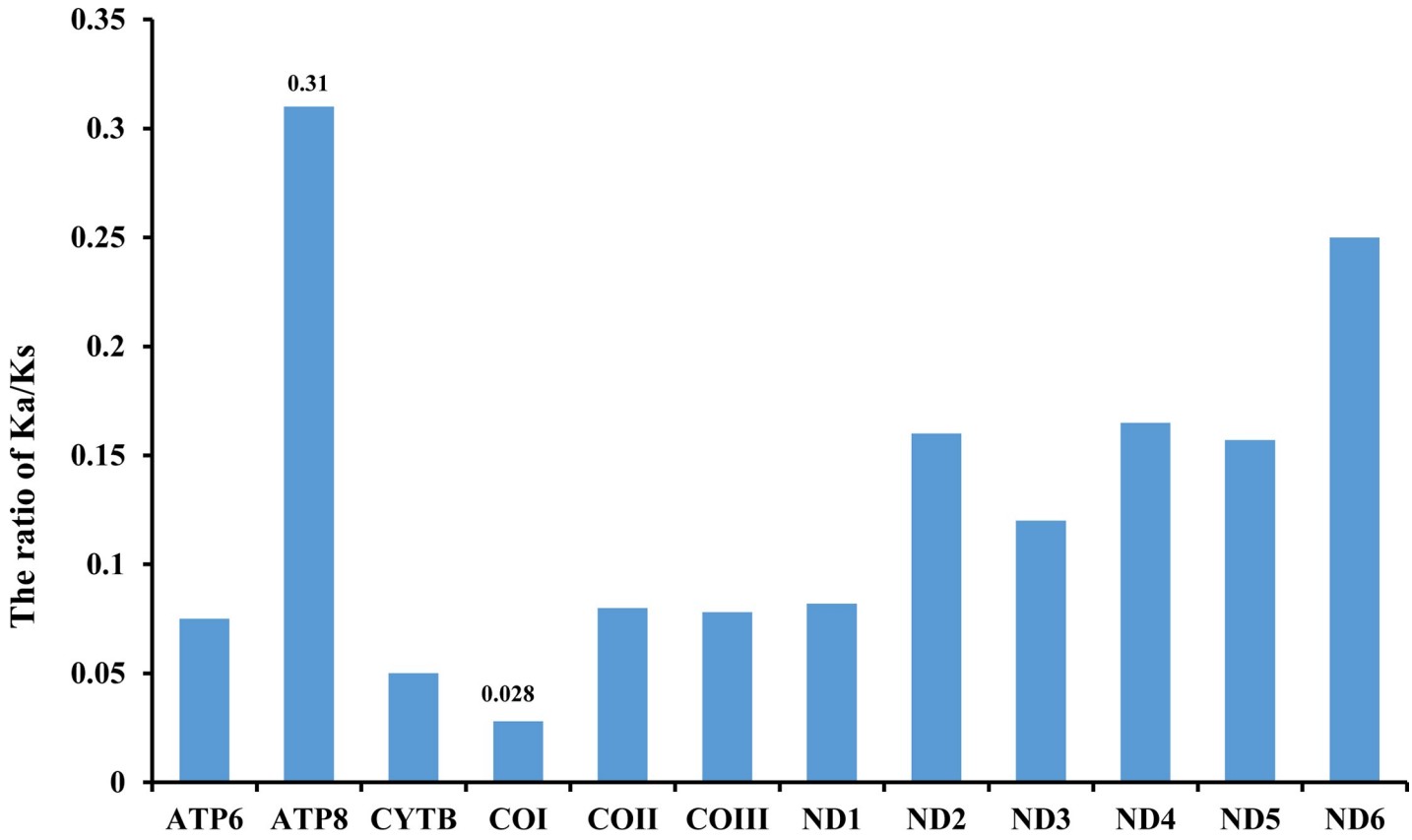

**Fig 6. Evolutionary rates of 13 PCGs in the mitogenomes of fifteen Mantodea species.** PCGs: protein-coding genes. The ratio of Ka/Ks (Ks: synonymous nucleotide substitutions; Ka: nonsynonymous nucleotide substitutions).

Amorphoscelidae, Mantidae, Deroplatyidae, Gonypetidae, Nanomantidae, and Metallyticidae. Four species of them (belonging to Deroplatyidae, Amorphoscelidae and Photinaidae) are from sequencing, while the others are from Genbank databases (Table 1). The samples were stored in 100% ethanol at −80˚C before use.

### 3.2. Mitochondrial genome sequencing, assembly, and annotation and analysis

Up to 2GB of raw reads were obtained individually from every sample and then trimmed of adapter contamination with the aid of NGS-Toolkit [43], while short reads and low standards were eliminated [44]. Velvet 1.2.10 was used to accumulate the fine readings according to the following criterion: Overlap Identity = 80–100 bp, Mismatches per Reading = 5%, Minimum Overlap = 30–50 bp, Maximum Gap Size = 3 bp [45]. The four complete mitogenomes were used to scrutinize the precision of the assembly.

Four overlapping nucleotide sequences were assembled by using DNAMAN software, which confirmed through manual inspection. In order to confirm the accuracy of the next-generation sequencing, we redesigned the primers to run PCR to confirm the sequencing results (S1 Table). Long-PCR reactions were run with the following cycling conditions: an initial denaturation for 2 min at 95˚C, followed by 35 cycles of 30 s at 92˚C, 30 s at 60˚C, 10 min at 72˚C, and final extension of 10 min at 68˚C using LA Taq (5 U/µ L, Takara). All PCR products were sequenced in both directions by the primer-walking method in Biotech Company.

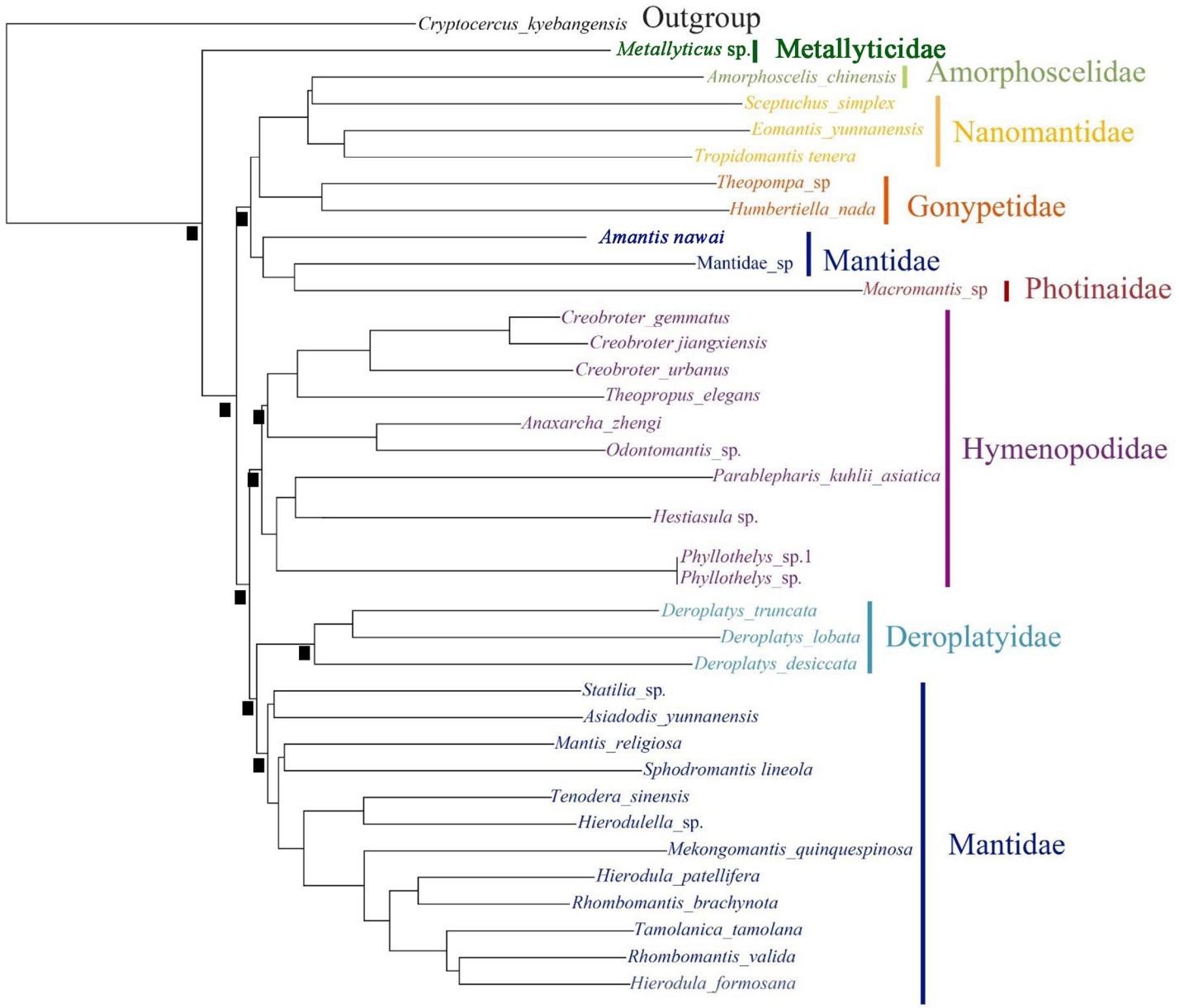

**Fig 7. Phylogenetic relationships of the Mantodea inferred from mitochondrial genome.** The black square implies that ML bootstrap support values and Bayesian posterior probabilities in percentages are greater than 90.

We also used ORF Finder and BLAST search engines against the GenBank database to identify protein-coding and rRNA genes. Furthermore, homologous genes from other Mantodea species were aligned with targets to further recognize these genes. ARWEN was used for the identification of transfer RNA genes by their cloverleaf secondary structure with standard parameters and tRNAscan-SE 1.21 with Search Mode = EufindtRNA-Cove, Genetic Code = Invertebrate Mito and Cove score cutoff = 0.1. MEGA 5 was used to analyze base composition [46–48]. Non-synonymous (dN) / synonymous (dS) mutation rate ratios of these four mitogenomes were calculated with DnaSP v5 among the 13 PCGs [49]. GenBank was used to recovered the sequences of conserved mt genomes of other Mantodea species.

Our data have been submitted to the NCBI database. In detail, The NCBI accession numbers are *D. truncate* (MT370514), *D. lobate* (MT370513), *A. chinensis* (MT370512) and *Macromantis* sp. (MT370515).

### 3.3. Sequence alignment and phylogenetic analysis

In our phylogenetic analysis a total of thirty-five insect species were involved (Table 1). One species of Blattaria, *Cryptocercus kyebangensis*, was chosen as an outgroup [26, 32]. Sequences of all mitochondrial protein-coding genes and rRNA genes except *nad4L*, *ATP6* and *ATP8* were exploited in the phylogenetic analysis. Nad4L is too short to align among the Mantodea species. *ATP8* was not identified in *D. truncate* and *D. lobate*. So nad4L and atp8 were excluded in the phylogenetic analysis. Alignment was made for phylogenetic analysis: a concatenated nucleotide sequence that includes two rRNA genes and ten protein-coding genes. MEGA 5 program used to execute the alignment of nucleotide sequences of all rRNA genes with protein-coding genes by following the standard criterion in ClustalW [46]. Poorly aligned sites were removed in the Gbocks server [50]. The Gblocks server ([http://molevol.cmima.csic.es/castresana/Gblocks_server.html](http://molevol.cmima.csic.es/castresana/Gblocks_server.html)) was applied with the 'protein' mode for PCG amino acid sequences, and with all options for a stringent selection were chosen.

Successive analyses were executed on the fused dataset using Bayesian inference (BI) and Maximum likelihood (ML), which were carried out with the aid of MrBayes 3.2 and RAxML 7.7.1, respectively. The GTRGAMMA model was chosen for the two datasets, with 1000 bootstrap replicas from the ML. The best-fitting nucleotides models were selected using Partition Finder V1.1.1 [51–53] from the BI as follows: TIM+I+G: *rrnL*, *rrnS*; GTR+I+G: *cox1*, *cob*, *cox2*, *cox3*, *nad1*, *nad5* and *nad3*; HKY+I+G: *nad6*; the best-fitting amino acids models were chosen as follows: MtArt+I+G+F: *nad1*, *nad4*, *nad5*; MtMam+I+G *cox1*; MtRev+I+G+F: *cox2*, *cox3*, *cob*, *nad3*, *nad6* and *nad4L*. Two individualistic sets of Markov chains were programmed, each of them having one cold and three heated chains for $1\times10^7$ generations, and every 1000th generation was evaluated. Convergence was concluded when a standard deviation of cleave frequencies <0.01 was accomplished. Sump and sumt burning arc were adjust to 25% and contype was adjust to all compat.

In summary, four species of Mantidae, belonging to Amorphoscelidae, Photinaidae, and Deroplatyidae families, were sequenced and annotated. These mitogenomes allocated the identical gene order and gene content alongside most of the known Mantodea mitogenomes. We demonstrated an inclusive comparative examines of the Mantodea mitogenomes and got the mitogenome characteristics and evolutionary patterns as an outcome. Most species presented alike usage bias in codons and nucleotides. The relatively variable and conserved regions were diversely diversified in the secondary structures of rRNAs and tRNAs. Also, based on the minimal classification system, phylogenetic analyses between 35 Mantodea species propose that the mitogenome is an effective marker for tagging family rank phylogenetic linkage among Mantodea.

## Supporting information

**S1 Fig. Live habitus images of *Deroplatys truncate*, *Amorphoscelis chinensis*, *Macromantis* sp. and *Deroplatys lobata*.** (A) *Deroplatys truncate* (B) *Amorphoscelis chinensis*; (C) *Macromantis* sp. (D) *Deroplatys lobata*.
(DOCX)

**S2 Fig. Inferred secondary structure of rrnS of *Deroplatys lobate*.** Inferred Watson-Crick bonds are illustrated by lines, whereas the noncanonical interactions are illustrated by dots.
(DOCX)

**S3 Fig. Inferred secondary structure of rrnL of *Deroplatys lobate*.** Inferred Watson-Crick bonds are illustrated by lines, whereas the noncanonical interactions are illustrated by dots.
(DOCX)

**S4 Fig. Inferred secondary structure of rrnS of *Deroplatys truncate*.** Inferred Watson-Crick bonds are illustrated by lines, whereas the noncanonical interactions are illustrated by dots.
(DOCX)

**S5 Fig. Inferred secondary structure of rrnL of *Deroplatys truncate*.** Inferred Watson-Crick bonds are illustrated by lines, whereas the noncanonical interactions are illustrated by dots.
(DOCX)

**S6 Fig. Inferred secondary structure of rrnS of *Macromantis* sp.** Inferred Watson-Crick bonds are illustrated by lines, whereas the noncanonical interactions are illustrated by dots.
(DOCX)

**S7 Fig. Inferred secondary structure of rrnL of *Macromantis* sp.** Inferred Watson-Crick bonds are illustrated by lines, whereas the noncanonical interactions are illustrated by dots.
(DOCX)

**S8 Fig. Inferred secondary structures of 18 transfer RNAs (tRNAs) identified in *Deroplatys lobata*.** Bars indicate Watson-Crick base pairings, and dots between G and U pairs mark canonical base pairings in RNA.
(DOCX)

**S9 Fig. Inferred secondary structures of 19 transfer RNAs (tRNAs) identified in *Deroplatys truncate*.** Bars indicate Watson-Crick base pairings, and dots between G and U pairs mark canonical base pairings in RNA.
(DOCX)

**S10 Fig. Inferred secondary structures of 17 transfer RNAs (tRNAs) identified in *Macromantis* sp.** Bars indicate Watson-Crick base pairings, and dots between G and U pairs mark canonical base pairings in RNA.
(DOCX)

**S1 Table. Verification PCR primers used for amplification of the mitochondrial genome of *Amorphoscelis chinensis*, *Deroplatys truncate*, *D. lobate* and Macromantis sp.**
(DOCX)

**S2 Table. Genes order and ORF features of the complete mitogenomes of *Deroplatys truncate*, *D. lobate*, *A. chinensis*, and Macromantis sp.**
(DOCX)

## Acknowledgments

We greatly thank Zeyi Lyu and Xianting Zhou provided the praying mantises specimens and ecology pictures.

## Author Contributions

**Conceptualization:** Yan Shi.

**Data curation:** Yan Shi, Lin-Yu Li.

**Formal analysis:** Yan Shi, Lin-Yu Li, Qin-Peng Liu, Zhong-Lin Yuan.

**Funding acquisition:** Yan Shi, Zhong-Lin Yuan, Tong-Xian Liu.

**Methodology:** Yan Shi, Qin-Peng Liu.

**Project administration:** Zhong-Lin Yuan, Guy Smagghe, Tong-Xian Liu.

**Resources:** Yan Shi, Qin-Peng Liu.

**Supervision:** Guy Smagghe, Tong-Xian Liu.

**Writing – original draft:** Yan Shi, Muhammad Yasir Ali, Guy Smagghe.

**Writing – review & editing:** Yan Shi, Muhammad Yasir Ali, Guy Smagghe, Tong-Xian Liu.

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
