## [Decision Letter · Decision Letter 0]

15 Apr 2021

PONE-D-21-03889

Complete mitochondrial genomes of four species of praying mantises (Dictyoptera, Mantidae) with ribosomal second structure, evolutionary and phylogenetic analyses

PLOS ONE

Dear Dr. Smagghe,

Thank you for submitting your manuscript to PLOS ONE. After careful consideration, we feel that it has merit but does not fully meet PLOS ONE’s publication criteria as it currently stands. Therefore, we invite you to submit a revised version of the manuscript that addresses the points raised during the review process.

One reviewer identify some minor editorial issues that would improve the grammar and presentation of the study and is worth addressing since this would improve the manuscript overall. The other reviewer identify a number of methodological issues, notably that some genes that was missing might be present in the genome. I suggest that the authors look at all of these methodological issues and address them since they will all improve the manuscript significantly and prevent unnecessary errors that might detract from the study.

We look forward to receiving your revised manuscript.

Kind regards,

Ben J Mans, PhD

Academic Editor

PLOS ONE

Journal Requirements:

In your Methods, please state the source of all insect samples used in your study. If any were collected directly from the field, please include details of the geographical provenance and any permits obtained for these collections.

We suggest you thoroughly copyedit your manuscript for language usage, spelling, and grammar. If you do not know anyone who can help you do this, you may wish to consider employing a professional scientific editing service. 

Please include your tables as part of your main manuscript and remove the individual files. Please note that supplementary tables (should remain/ be uploaded) as separate "supporting information" files

In your Data Availability statement, you have not specified where the minimal data set underlying the results described in your manuscript can be found. PLOS defines a study's minimal data set as the underlying data used to reach the conclusions drawn in the manuscript and any additional data required to replicate the reported study findings in their entirety. All PLOS journals require that the minimal data set be made fully available. For more information about our data policy, please see http://journals.plos.org/plosone/s/data-availability.

Reviewers' comments:

Reviewer's Responses to Questions

**Comments to the Author**

1. Is the manuscript technically sound, and do the data support the conclusions?

Reviewer #1: Yes

Reviewer #2: Partly

2. Has the statistical analysis been performed appropriately and rigorously? 

Reviewer #1: Yes

Reviewer #2: I Don't Know

3. Have the authors made all data underlying the findings in their manuscript fully available?

Reviewer #1: Yes

Reviewer #2: Yes

4. Is the manuscript presented in an intelligible fashion and written in standard English?

Reviewer #1: No

Reviewer #2: No

5. Review Comments to the Author

Reviewer #1: In the manuscript titled 'Complete mitochondrial genomes of four species of praying mantises (Dictyoptera,

Mantidae) with ribosomal second structure, evolutionary and phylogenetic analyses' the author present the complete mitochondrial genome of four species of mantid. Overall the study is good and should be published but some revisions need to be addressed before publishing.

1) The authors need to have a native English speaker review the document for grammar. It is not unintelligible but there are many spots where it is difficult to follow or does not sound scientific.

2) Throughout the document, the species D. truncata and D. lobata are converted to truncate and lobate due to autcorrect from their word processor (it actually just did it now in this response portal). It is annoying but an easy fix.

3) I would like to see a table or tables that list the gene order and position for all ORFs for each species, this generally is a useful reference for other readers.

Reviewer #2: The authors sequence and annotate the mitogenomes of four mantises species and combine this new data with existing mantis data to estimate phylogenetic relationships in the group. They find a few missing genes in some of the four new mitogenomes, but tRNA secondary structure and codon usage was similar to other mantises. The phylogeny also tended to agree with current taxonomy.

I think the study is definitely a worthwhile addition to insect mitogenomics. I particularly appreciate that you discuss your findings in the context of insect and mantis mitogenomics. However, I do have several issues that should be addressed, some more major then others:

1. You need to re-work some of the structure in the Introduction. For example, the second paragraph introduced mitogenomes and then abruptly switches to introducing the taxonomy of mantises. I would separate these.

2. Because the Methods comes last, I would include some brief methodological details in the Results and Discussion section.

3. Please include more specific results. For example, you vaguely state that there is a high A+T% and list the most-used codons. Please provide specific values in the text.

4. You begin discussing the utility of using cox1 as a barcoding gene (starting line 151), but this discussion appears abruptly. Why is this relevant? Is it related to your nucleotide diversity results? Please explain.

5. I’m a little confused by your sampling strategy. Did you sample the 35 mantis species mentioned in lines 189-190? Or were these taken from GenBank?

6. I am not convinced you are actually missing the atp8 genes. In my experience, it can be difficult to annotate this gene, and you often have to look for it manually based on ORFs, etc. It also seems like there is a non-coding region on the 5’ end of your atp6 genes when a “missing” atp8 is reported. I would bet that is the atp8 gene. Also, were your missing genes confirmed with your PCR step? Finally, it looks like both atp6 and atp8 are missing from one species (bottom left in Fig 1), but I didn’t see any discussion of this. Please elaborate.

7. Did you confirm that these assembled mitogenomes are in fact complete and circular? I recommend doing this, especially since you report missing genes.

8. Please provide more methodological details. For example, how did you calculate Ka/KS? What were your PCR conditions? What GBlocks parameters did you use?

9. There are several portions of the manuscript that are difficult to read. I recommend going through the text and again and edit for grammar, awkward phrasing, etc.

Some minor comments:

Line 25: There is no context for referencing the “Schwarz and Roy (2019) taxonomic system.” Perhaps just say “the latest taxonomy.”

Line 31: No need to say “respectively.” Same in line 82.

Line 42: Be careful about using “bugs” when you mean “insects.” Entomologists will strongly object to this usage…

Lines 74-78 should be in the Introduction

Line 100: Why do you rule out this second possibility? Please elaborate.

Line 144: Please provide a brief overview of dN/dS ratio (i.e., dN/dS > 1 is evidence for positive selection, < 1 for purifying selection, and =1 for neutral).

Line 217: Confusing phrasing. Please re-write. It’s not clear what “miniscule to align means.”

Figure 1: It’s not clear which mitogenomes go with each species. Please add names to the figure.

6. PLOS authors have the option to publish the peer review history of their article (what does this mean?). If published, this will include your full peer review and any attached files.

Reviewer #1: No

Reviewer #2: No

---

## [Author Response · Author response to Decision Letter 0]

24 May 2021

4/5/2021

Dear Editor, 

Thank you very much for your email message regarding our manuscript. We appreciated much the valuable comments from the reviewers for revising and improving our manuscript. Generally, we have incorporated all suggestions and comments of yours into this revision. More specifically, our corrections are as follows here under.

We hope you can now accept this revised version. 

Most sincerely,

All authors

Reviewer #1: In the manuscript titled 'Complete mitochondrial genomes of four species of praying mantises (Dictyoptera, Mantidae) with ribosomal second structure, evolutionary and phylogenetic analyses' the author present the complete mitochondrial genome of four species of mantid. Overall the study is good and should be published but some revisions need to be addressed before publishing.

1) The authors need to have a native English speaker review the document for grammar. It is not unintelligible but there are many spots where it is difficult to follow or does not sound scientific.

Reply: We thank this referee for his/her suggestions. We have polished the full text and further revised the corresponding parts. 

2) Throughout the document, the species D. truncata and D. lobata are converted to truncate and lobate due to autcorrect from their word processor (it actually just did it now in this response portal). It is annoying but an easy fix.

Reply: Thank you. 

3) I would like to see a table or tables that list the gene order and position for all ORFs for each species, this generally is a useful reference for other readers.

Reply: We agree and have added the related information on the position of all ORFs. See Table S2. For the gene order, we refer to Figure 1 with the mitochondrial gene structure.

Reviewer #2: The authors sequence and annotate the mitogenomes of four mantises species and combine this new data with existing mantis data to estimate phylogenetic relationships in the group. They find a few missing genes in some of the four new mitogenomes, but tRNA secondary structure and codon usage was similar to other mantises. The phylogeny also tended to agree with current taxonomy.

I think the study is definitely a worthwhile addition to insect mitogenomics. I particularly appreciate that you discuss your findings in the context of insect and mantis mitogenomics. However, I do have several issues that should be addressed, some more major then others:

1. You need to re-work some of the structure in the Introduction. For example, the second paragraph introduced mitogenomes and then abruptly switches to introducing the taxonomy of mantises. I would separate these.

Reply: Agree. We have separated the two parts. Please check the line 53.

2. Because the Methods comes last, I would include some brief methodological details in the Results and Discussion section.

Reply: Thank you. We have added the brief methodological details in the Results and Discussion section. Please check lines 78-83. 

3. Please include more specific results. For example, you vaguely state that there is a high A+T% and list the most-used codons. Please provide specific values in the text.

Reply: Agree. We have added the whole genome, A+T-rich region and PCGs specific values of four mitochondrial genomes. Please check Table S2.

4. You begin discussing the utility of using cox1 as a barcoding gene (starting line 151), but this discussion appears abruptly. Why is this relevant? Is it related to your nucleotide diversity results? Please explain.

Reply: Thank you. The dN/dS pairwise analysis showed that cox1 is undergoing a robust purifying selection (0.028). That is why cox1 can be used as a molecular marker of barcoding entire Mantidae. We also reorganized the writing of this part. Please check lines 152-153.

5. I’m a little confused by your sampling strategy. Did you sample the 35 mantis species mentioned in lines 189-190? Or were these taken from GenBank?

Reply: We have reorganized the writing of this part. Please check lines 189-193 and Table 1.

6. I am not convinced you are actually missing the atp8 genes. In my experience, it can be difficult to annotate this gene, and you often have to look for it manually based on ORFs, etc. It also seems like there is a non-coding region on the 5’ end of your atp6 genes when a “missing” atp8 is reported. I would bet that is the atp8 gene. Also, were your missing genes confirmed with your PCR step? Finally, it looks like both atp6 and atp8 are missing from one species (bottom left in Fig 1), but I didn’t see any discussion of this. Please elaborate.

Reply: Thank you for this advice. We know ATP8 which is difficult to annotate. We repeated to analysis the sequence for annotation of ATP8, and in addition we ran the PCR to amplify the corresponding part of about 1000bp to confirm the sequence. Unfortunately, we could not annotate the relevant homologous sequences. For the ATP6 of Macromantis sp. species, we have added this annotation name in Figure 1 (bottom left in Fig 1) and the Table S2. Please check Figure 1 and Table S2. 

7. Did you confirm that these assembled mitogenomes are in fact complete and circular? I recommend doing this, especially since you report missing genes.

Reply: We thank this referee for his/her comment. We received the assembled mitogenomes sequence and this was confirmed by PCR technology. The starting and the ending of sequence overlapped with 184bp, so we are sure that this is a complete and circular structure. We redesigned the primers for PCR and sequenced the PCR product to confirm the sequencing results. The primers are listed in Table S1. We also used ORF Finder and BLAST search engines against the GenBank database to identify protein-coding and rRNA genes. Please check the lines 206-208.

8. Please provide more methodological details. For example, how did you calculate Ka/KS? What were your PCR conditions? What GBlocks parameters did you use?

Reply: We agree and have added the methods of calculating Ka/KS. Please check lines 219-221. We also added the conditions of long PCR reactions. Please check lines 213-217. Also the GBlocks parameters were added in lines 237-239.

9. There are several portions of the manuscript that are difficult to read. I recommend going through the text and again and edit for grammar, awkward phrasing, etc.

Reply: Thanks for your suggestion. We polished the text and revised the corresponding parts. 

Some minor comments:

Line 25: There is no context for referencing the “Schwarz and Roy (2019) taxonomic system.” Perhaps just say “the latest taxonomy.”

Reply: Done; please check lines 26-27

Line 31: No need to say “respectively.” Same in line 82.

Reply: Done; please check lines 31, 84

Line 42: Be careful about using “bugs” when you mean “insects.” Entomologists will strongly object to this usage…

Reply: Done, please check line 42.

Lines 74-78 should be in the Introduction

Reply: Agree. We have moved this text to the Introduction. Please check line 64-68.

Line 100: Why do you rule out this second possibility? Please elaborate.

Reply: We apologize for this. Indeed the second possibility is also very likely. So, we have reorganized this text part. We have deleted our conclusion: “We concluded that a lack of the atp8 gene in D. truncate and D. lobate is most likely because of the former or the later reason.” Please check lines 112-113. Thank you.

Line 144: Please provide a brief overview of dN/dS ratio (i.e., dN/dS > 1 is evidence for positive selection, < 1 for purifying selection, and =1 for neutral).

Reply: Agree. Please check lines 151-152.

Line 217: Confusing phrasing. Please re-write. It’s not clear what “miniscule to align means.”

Reply: We thank this referee and have rephrased this part text, which means Nad4L is too short to align among the Mantodea species. ATP8 was not identified in D. truncate and D. lobate. So nad4L and atp8 were excluded in the phylogenetic analysis. Please check lines 238-241. 

Figure 1: It’s not clear which mitogenomes go with each species. Please add names to the figure.

Reply: Thanks and done.

---

## [Decision Letter · Decision Letter 1]

25 Jun 2021

PONE-D-21-03889R1

Complete mitochondrial genomes of four species of praying mantises (Dictyoptera, Mantidae) with ribosomal second structure, evolutionary and phylogenetic analyses

PLOS ONE

Dear Dr. Smagghe,

Thank you for submitting your manuscript to PLOS ONE. After careful consideration, we feel that it has merit but does not fully meet PLOS ONE’s publication criteria as it currently stands. Therefore, we invite you to submit a revised version of the manuscript that addresses the points raised during the review process.

Minor issues were raised which can be considered since they will improve the manuscript somewhat.

We look forward to receiving your revised manuscript.

Kind regards,

Ben J Mans, PhD

Academic Editor

PLOS ONE

Journal Requirements:

Reviewers' comments:

Reviewer's Responses to Questions

**Comments to the Author**

1. If the authors have adequately addressed your comments raised in a previous round of review and you feel that this manuscript is now acceptable for publication, you may indicate that here to bypass the “Comments to the Author” section, enter your conflict of interest statement in the “Confidential to Editor” section, and submit your "Accept" recommendation.

Reviewer #2: (No Response)

2. Is the manuscript technically sound, and do the data support the conclusions?

Reviewer #2: Yes

3. Has the statistical analysis been performed appropriately and rigorously? 

Reviewer #2: Yes

4. Have the authors made all data underlying the findings in their manuscript fully available?

Reviewer #2: Yes

5. Is the manuscript presented in an intelligible fashion and written in standard English?

Reviewer #2: Yes

6. Review Comments to the Author

Reviewer #2: Thank you for your thorough responses and edits to your manuscript based on my comments. I think the manuscript has been improved. I do have a few comments that still should be addressed, based on my first round of comments:

1. I still think the paragraphs in the Introduction are a little disconnected. Perhaps switch paragraph #2 (on the utility of mitogenomes for mantid phylogenetics) and paragraph #3 (on mantid taxonomy).

2. I think you should add the AT% to the main text, not just in a supplement.

3. I’m still somewhat skeptical that the atp8 gene is actually missing, but your combination of NGS and PCR results are more convincing. It is difficult for me to fully evaluate this claim without access to the sequence data; but as is stated in the manuscript, atp8 and other genes are missing from other taxa. Is this the first case of missing mt genes in mantises?

4. I recommend one more round of editing for writing clarity. For example, Lines 50-51 are a little confusing, and should probably be split up into two sentences.

7. PLOS authors have the option to publish the peer review history of their article (what does this mean?). If published, this will include your full peer review and any attached files.

Reviewer #2: No

---

## [Author Response · Author response to Decision Letter 1]

27 Jun 2021

26/06/2021

Dear Editor, 

Thank you very much for your email message regarding our manuscript entitled “Complete mitochondrial genomes of four species of praying mantises (Dictyoptera, Mantidae) with ribosomal second structure, evolutionary and phylogenetic analyses”. 

We appreciated the valuable comments from the reviewers and editor to revise and improving our manuscript. Generally, we have incorporated all suggestions and comments into this revision. More specifically, our adjustments are stated below in a point-by-point manner.

We hope that this revised version can be accepted for publication. 

Most sincerely,

All authors

Reviewer #2: 

Thank you for your thorough responses and edits to your manuscript based on my comments. I think the manuscript has been improved. I do have a few comments that still should be addressed, based on my first round of comments:

1. I still think the paragraphs in the Introduction are a little disconnected. Perhaps switch paragraph #2 (on the utility of mitogenomes for mantid phylogenetics) and paragraph #3 (on mantid taxonomy).

Reply: We agree and have switched the paragraph #2 and paragraph #3. See lines 46-68.

2. I think you should add the AT% to the main text, not just in a supplement.

Reply: Indeed, we have added the AT% information to the main text; see in Table 2.

3. I’m still somewhat skeptical that the atp8 gene is actually missing, but your combination of NGS and PCR results are more convincing. It is difficult for me to fully evaluate this claim without access to the sequence data; but as is stated in the manuscript, atp8 and other genes are missing from other taxa. Is this the first case of missing mt genes in mantises?

Reply: Thanks for your suggestion. Here we attach mitochondrial sequencing results for D. truncate species, and we really did not identify the atp8 gene. So, we imply that atp8 gene may have been lost. Yes, from our sequence of D. truncate and D. lobata, we can confirm the atp8 is lost. This is the first case of missing Mt gene in mantises. 

TAATTATCCCATGAAAGATTAGTATTAAACCAATAACTCTAAATTTATTTTAAATAATAGAGTAAAATGCCTGATTTAAAAGGAATATCATGATAGGATAAAATATGTAAATTAATTACTTTTACTAATATTATCATTACACATAACAGAGTTAAACTGTTTCCTAAAATATCAAAAATTTTTGTGCATCCTACACTAAAATATAGAAAAGTAAGCTAAATTTAAGCTAATGGGTTCATACCCCATATATAGAGTTTACACTCTCTTTTCTAGTGCCCAACAATTCGATAAAAATCCTCTTTTTAATAACTTTAATTAGAGGTGTGTTAATTTCATTATGTGCTAATTCATGAATAGGAGCATGAATAGGATTAGAAATTAACTTACTCTCCTTCATTCCTTTACTTTCTTCCAGAAAAAACTTGTTCTCAACTGAAGCTTCACTTAAATATTTTCTTATTCAAGCTATTTCATCATCTACATTTCTATTTCTAATTATGATAAAAATTAATATCCAAGAAATGTTTTATCTAATAAAATTTAATAATTGAAACACCTTAATTACAATTCCTATTCTAATGAAAATTGCATCTGCACCTTTTCATTGATGATTACCTTCCGTGATAGAAGGTTTATCTTGAATAAATTGTTTTATTATTCTGTCAATTCAAAAAGTAGCCCCATTAATATTCATTTCCTACTTAATCACTAATAACTTCTTTATTCAAATCATCATTACAATTTCTGCAATTATAGGAGCTATTGGAGGTCTCAATCAAATTTCTCTACGAAAAATTTTATCATTTTCATCTATCAACCATATTGGATGAATATTAACAACCATAATTATAGGTTCTAACTTTTGAATAACATACTTTATCATTTATACGGTAAATATTATCCCTATTATTTTTATGGTAAACAAAATAAACCTATCTTTTATTCCTCAGACCTTTAATTCCTTCAATAACAAAAAAATTATTAAATTCATTCTATTTATCTCCCTCCTTTCTTTAGGGGGCCTCCCTCCTTTTATTGGTTTTTTCCCAAAATGAATTATCATTCAAATTATAATCCAAAATTTAATAATTCTTACATCAATAACCCTAATTATATCCTCCCTTTTAACCCTTTATTATTATCTACGAATTATTTACACAACATTAATAATCACAAATTCAGAAATAACTTGAATAGCAATCTACTCAAATAACAACTTAGGAAAAAGAACATTTCTTTTCTTATTCATTCTTATTTTTGGTTTATCAATCTGTACACTTATCTTAACAATCTATTAAGATTTTAAGTTAACCAAACTAATATCCTTCAAAGTTATAAATAAAGTGATTCATCTTTAAGTCTTAGTATTACTTACACCTTTAGAATTGCATTCTAATATCATCTCCATGAATATAAAACTTTAGTAAAAGAGATAATAATCTCATTAATAAATTTACAATTTATTACCTAATATCAGCCATTTTACTTTTTTTTTTGCAACGATGATTATTCTCAACAAATCATAAGGATATTGGAACATTATATTTTATTTTTGGTGCATGGGCAAGTATATTAGGAACATCTTTAAGAATTTTAATTCGAACAGAGTTAAGACAACCCGGATCTTTAATTGGAGATGATCAAATTTATAATGTCATTGTAACAGCTCACGCTTTTATTATAATCTTCTTTATAGTTATACCCATTATAATTGGTGGATTTGGTAATTGATTAATTCCTTTAATACTTGGAGCTCCAGATATAGCCTTTCCTCGAATAAATAATATAAGATTTTGATTATTACCTCCTTCTATTCTATTTCTTATTATCGGAAGAGTTGTAGAAAGAGGAGCTGGTACAGGATGAACAGTCTATCCCCCTCTTTCAGCCAGAATTGCTCACGCCGGCCCCGCAATTGATTTAACCATTTTTTCCTTACACTTAGCTGGAATATCAAGAATTATAGGAGCTGTAAATTTTATCACAACTATAATTAACATAAAATCAATTCATATAAATCATACTCAAATTCCTTTATTTGTTTGATCTGTAGGAATTACAGCAATTCTTCTTCTTTTATCTCTCCCTGTACTTGCTGGGGCAATCACTATACTTCTAACTGATCGAAACTTAAATACATCCTTTTTCGACCCTGCTGGAGGGGGGGATCCTATTCTCTATCAACATCTATTTTGATTTTTTGGACATCCTGAAGTATATATTTTAATTTTACCAGGATTTGGTATAATTTCTCATGTCATTTCTCATGAAAGAGGAAAAAAAGAAGCTTTTGGAAATTATGGAATAATTTGAGCTATATCAGCCATTGGTTTTCTGGGGTTTATTGTATGAGCTCATCATATATTTACAGTAGGAATAGATGTAGATACACGAGCCTACTTTACAGGAGCTACTATAATTATTGCTGTTCCTACGGGTATTAAAATTTTTAGTTGACTCACAACTATATACGGAACAAAAATAATTTATAGTGTAGTATCTTTATGAGCATTAGGATTTATTTTTTTATTTACAGTTGGAGGTCTTACAGGAGTAATTTTAGCCAATTCAGCTATTGATATTATTCTTCATGACACTTATTATGTAGTAGCCCATTTTCACTATGTACTTTCAATAGGAGCCGTTTTTGCTATTATAGCAGGGTTCATTCATTGATATCCCTTATTCACTGGATTATCATTAAATCCTAATTGATTAAAAAGTCAATTTTTCACAATATTTGTAGGAGTTAATTTAACATTTTTTCCACAACATTTCTTAGGATTAGCTGGAATACCTCGACGCTATTCAGATTACCCTGACGCCTATAGATCATGAAACTTCTTATCATCTGTAGGAGCAATAATTTCTTTCGCTGCTGTTATCATATTCATTCTTATCTTATGAGAAAGAATTACCTTAAATCGATTTATATTATTTTCATCTCAAATAAATAGATCAATTGAATGAATTCATAATTTTCCCCCAGCTGAACACACCTACAATGAACTAACTCTAATTACAAATTAAAACCTAAATTGTTGATAATTTTCCTCATTTCTAATGTGGCAGAATAGTGCACTGGGTTTAAGCTCCAAAAATAAAGATAAACTTTTTTTAGAAGTTATTTTAATGGCTACAAATGCAAATTTAGGATTTCAAGATAGAGCCTCCCCCTTAATAGAACAACTCATGTATTTTCATGATCATTCCATATTTATTATTACTATAATTGTAATTACAGTAAGTTATATAATTATAGCTTTAATAGTAAATAAATTTTCTGATCGTCATGTTATAGATGGTCAATATTTAGAAATTTTTTGAACAGTTCTTCCAGCCATAGTTCTAGTCTTCATCGCTCTACCTTCTCTACGGATTTTATACCTAATTGATGAAAATACAAATCCAACATTAACCTTAAAAACAATTGGTCATCAATGGTATTGAAGTTACGAATATTCAGATTTTACCAATATTGAATTTAACTCTTACATAATTCCACAAAATGATTTAAATCTATTTAACATGCGTTCACTTGAAGTTGACAATCGAACATCATTACCCATAAATACCTTAACACGAATTTTAATCACCTCAGATGATGTTATTCATTCTTGAACAATTCCGAGAATTGGAGTAAAAGCTGATGCTACTCCGGGACGATTAAATCAAGCAACATTTTGATTTAATCGTCCCGGAGTATTTTATGGTCAATGTTCAGAAATTTGTGGAGCAAATCACAGATTTATACCTATTGTAATTGAAAGAACTTTGATTAATAATTTTCTTAGTTGAATCTTAAATTATATTGAATCACTAGATGACTGAAAATAAGTGATGGTCTCTTAAACCATATCATAGTAACATAATAACTACTTCTAGTGATTGACTAACAATTTATCAAGAAGTTAGTTAAAAAATAACATTAGTATGTCAAACTAAAGTCATTATCATTTAATACATCTTTATACCCCAAATAATACCCCTAAATTGGCTAATCCTATTTTACATTGTTTCTACTAGATTAATTTTTTTTAATGTAATAAATTTTTTTATATTTTCCCACAAAATCCCATTAACTTCAAATAAAATTTTACTAAAAACCCTAATTTGAAAATGATAACCAACCTATTTTCAATTTTTGATCCCTCTTCAAATTTTATAAACCTATCAATAAATTGACTTAGAATTTGAATCGGATTATTATTATTTCCTTCCTCGATATGGTTAATTTCATCACGAAATAAAACCCTTTGAAGTTTTATTTTAAGTAAACTTCATGAAGAATTTAAATTATTAATTGGTAAAAAAAAAATTAACAAAGGATCAACATTCATATTTATTTCAACATTTTTACTTATTATATATAATAATTGTATAGGATTATTTCCATATATTTTTACTGGGACAAGTCATATAGCTATAACTCTATCTTTTGCTTTACCTTTATGACTAAGATTTATACTCTTTGGTTGAATTAATAACTCTAATCACATATTTATTCATTTAGTTCCTCAAGGAACTCCAAATATATTAATACCTCTTATAGTTTGTATTGAAACAATTAGAAATTTAATTCGTCCCGGAACTTTAGCTATTCGACTCGCAGCAAATATAATTGCAGGACATTTATTAATAACCCTCCTAGGAAATTCTGGGAGAAACATTATAGATTCATTTTTACCCCTATTAATTTTAGTTCAAATTATACTTTTAACTCTAGAATCCGCAGTTGCTATTATCCAATCATATGTTTTTGCAGTATTAAGTACTTTATACTCTAGAGAAGTAAATTAATAATGATAATACACACTAATCACCCCTACCATTTAGTTACTTATAGTCCCTGACCTATTATAACTACTTTAAGAATCATAATCATAATATTAGGTTTTATTAAATTTTCTTATGAGTTTAGTGAAAAATTTATGCTATTAGGAACTTTAATTTTAATTTTAATTACTACTCAATGATGACGTGACGTTGTACGAGAAAGTACATATCAAGGATTACACACTAAAAAAGTAATCTTTGGACTACGATGAGGAATAATTTTATTTATTATTTCAGAAATTTTTTTCTTTGTATCTTTTTTTTGAACTTTTTATCATAGAAGCTTAACTCCTACTATTGAATTAGGGTCCTTTTGACCACCTCAAGGAATCTGACCCTTTAATGCTCTTCATGTTCCTCTTCTTAATACAACGGTACTTTTAGCATCAGGCATCACTATTACATGAAGTCACCATGGACTATTAATAAATAATTATAATCAAGCCACCCAAGGATTAATATTTACCATTATCCTTGGGATTTATTTTACCATCTTACAACTCTATGAATATTATGAAGCTCCGTTTACAATTGCGGATTCAGTTTTTGGGTCAATCTTTTTTATAGCAACTGGATTTCATGGACTTCATGTAATTATTGGAACTACATTCTTAGTTACATGCTTATTTCGAATAATTTATAAACATTTTACATCTATTCACCACTTTGGTTTCGAGGCAGCAGCCTGATATTGACATTTTGTGGACGTAGTATGATTATTCCTGTACATTTCTATTTATTGATGAGGGAGATAAATCCAATTTATTTAGTATAAAAGTACAATTGATTTCCAATCAAAAAGTCTATATTTAATTAGAATAAATAATTAAAATTTTAATTTTTATCTCATTCATTACTATATCAATTACCTTAACAATCATATTACTAACAAATTTCTTGTCAAAGAAAAAAATTGAAGACCGAGAAAAAAATTCACCTTTTGAATGTGGATTTGATCCGATTAGATCCTCGCGCCTTCCCTTTTCTTTACGTTTCTTTTTGATTGGAGTAATCTTTTTAATTTTTGATGTAGAAATCGCCTTTATCTTACCAATAATTATCATTCCTCTTACATCAAAAATAACATCTTGAATATCTACTAGAATTATATTCTTATTGATCTTAACAACTGGTTTATTTCATGAATGAAATCAAGGTTCTCTCGACTGAGCAACTTAACTTTATAAGGGTTATAGTTAAAAATAACATTTGACTTGCACTCAAAAAGTATTGAAATATCAATTTTCCTTATTATAAGTAAGAAGCAAATTCATTGTAATCAGTTTCGACCTGATAGTAAGATATTCATATCCTTATTTGTTGATTAATTGAAACCAAATAGAGGTATATCACTGTTAATGGTAAAATTGAAATTAATGCTTTCCAATTAAGAAAATGTGTAGATCGAATATAAGTTGCTAATTTATTATTCAAGTGGTTTAATCCCATTTACATTTTAATTTATATAGTTTAAATAAAACATTACATTTTCATTGTAAAAATAAATTTTTCAATTTTATTAATAGTAAGTACTCTATTTAAAGATAAATTAGTTATCTCAATAACAGCTTCAATGTTATACTCTCTATAAGATATTTAAATAAATACAAAATATTATTATAAGTAAAATACTAAAAAAAATCATTAAATAAGATTTTAAATCATTATATTGAAATCATTGATTAAAACATCTTAATTTTATCAAAATATAATATAAATTTTGTGCTCCAAAATATTCTCTTCAGCCTAAATCAATATATATCATTGAATTTAAACTAATAATTAAAGGTATTTTTCTTACTCCTTTAGTAAAAATTAAAGGTATATATCATATTGAAGCAAAAAAAATTGTCATTTTATAATACTTAAATGTTGAAAAATAATAATTTATTCTATACTTAAATATAATTCTTCCTAATCATAAACCTACAAATCTACCTAATAAAGGTATTATTTTCATTATTAATGTCATATAAATTATATAGGGAGTAAGAAAAATAATCCAATTTAATATCCCACCTCCAATAATTGCAAATAATATTAAACCTAATATTCCATATACTATTATTCAACTTTCATTTAATTTACATATTGATATTATATTAAAATCTCCTCATAATACATAATAGGATAAACGAAAAGAATAACTTACTGTTAAACCTGTTGAAAAAAAAAATAAAACATATATAAAGATATTTAAATTTCTTAAAGATAATATTTCTAAAATTATATCTTTTGAATAAAATCCTGCTAAAAAAGGTATACCACATAAAGCAAAATTTGATACTATAAAACAAGATGAAGTAAAAGGCATAAATATTACTAAATTTCCTATAAAACGAATATCCTGTAAATTTTTTATTCTATGGATTATTATTCCTGTACACATAAATAACAATGCCTTGAATAAAGCATGAGTTAATAAATGAAAAAAAGCCAAATCTGAAAAACCTAATGATAAAATTCTTATTATTAATCCTAATTGACTTAAAGTTGATAAAGCAACAATCTTTTTTAAATCATATTCAAAATTTGCTCCTAACCCTGATATAAACATTGTTAAAACAGATATTACTAGTAAAAACTTTAATAATCAATCCGGAAAAACCTTACAAAATCGGATTAATAAATAAACCCCAGCCGCAACCAAAGTTGATGAATGAACTAGGGCTGAGACAGGAGTTGGAGCAGCTATTGCTGCAGGCAATCAAGCTGAAAAGGGAATTTGTGCACTTTTTGTTATTCCTGCTAAAAGTACTAAAAATGAAATTAAATATATTTCAATTTCATTTAATGTACAATCTAAATAAAAAATATAATTTCATCTTCCAAAATTTAATATTCATGAAATTGCTATTAATAAAGCAACGTCCCCTACTCGATTAGATAAAGCTGTCAATATTCCAGCATTATAAGACTTAGTATTTTGATAATAAATTACAAGACAATAAGAAATTAAACCTAACCCATCTCAACCTAATAAAATTCTAATCAAATTTGGACTAATAATTAAAAACATTATAGACAATACAAATATTAATACTAAAAAAATAAACCGATTTAGTGATGAATCTCCAGTTATATAATCTTCTCTATATAAAATTACTAAGGATGAAATTAATAAAACAAAACTCATAAATATGAGAGACATTCAATCCAACAATATTGTTATAACAACTGAAGAAGTACTTAATCTAACAATTTCTCATTCAATAAAAATAATTAAATCATTTAAAATAAAAACCATTCTAGAAATAAATATTATTAAAGAAAATAATGCTAATAAAAAAAATCTAATAAAACATAATGATAAATAAACCATAACTTAAAATAACTATTCATTATATCCATGATACCACAAATCATAATTTTATGATAAACTATTTAAGTAAATTAACTTTTATTTAAAATCAAATTAAAACATAATCTCTCTTTAAAATTAATAAATTTAAGGGTAATCAATGAAGTATTATTAATAAATATTCACGTCTATACCCTCCAACTCTTCTATAAATGCCTGAATAATAAAATCCATGTTGACTATATGAATATAAATATAAAGTATAAGCAGCTCTAAAAAATGAAATTAATATAAGAAAAAATATAGATATTCATACCCAACTTACTATACTATTGAATAATCCAATTTCTCCTAATAAATTTAAAGTAGGGGGAGCAGCTATATTGCTAGATGAAAGTAAAAATCATCATAATGTTAAACTTGGCATCAAATTTAATAAACCCTTATTAATTAACAATCTTCGTCTTCCCAGACGCTCATAACTAATATTAGCTAAACAAAATAAACCAGATGAACATAAACCATGAGCAATTATTATTACAAAAGTTATATAAAACCCTCAAATATTTAATGTTATTAAGCCCCCAATAATTAAACCTATATGAACCACAGACGAATAGGCAATTAATGCTTTTACGTCTATTTGACGTAAACACATTAAACTAACTATAACCCCACCGATTAATCTAGTAGAAAATCAAAATACATTAACTAATATCCCAACTCTCTTCAAGAATTCATATACTCGTAATAAACCATAACCTCCTAACTTTAATAAAACTCCAGCTAAAATCATTGAACCTGAAACTGGAGCTTCAACATGAGCCTTAGGTAATCATAAATGCACTAAAAATATAGGCATCTTAATCAAAAAAGCTAAAATTATTCTAAAATATAAATATCAATTCAATAAATTATTTTTATAAACTAACGAAAAAACTAAATAGCCCATGTTGTTATAAAAATATATAATACCCATCAATAATGGTAATGAAGCTAAAAGAGTGTATAATAATAAATAAATACCAGCTTGCAATCGCTCAGGTTGATATCCTCACCCAAAAATTAAAAATAAAGTTGGAATCAAACTACCCTCAAAAAAAAAATAAAATGAAATGAAATTTATTCTACAAAAAGTACAAATTAATATCAATAACAATATTAAAATTATAAATATAAATAAATTATTATAAAATTTATATCGAATAACAGAATAACTAGCTAAGATTATTAAAACACAAATTCAAAATCTTAGCACAATAAGACCATATGATAAGTAATCATATCCAAATATATAACTCATTCTTATTCAATAAAAAAAATCCACTCCAATAAAAAATTTAAAAGATATAAAAAATAATAAATTCTGAATTAATAATCAATTTTTATTCATTAAACATAATGGAATCAAAAACATTAAACTTAAAATATATCTTAACATTGCAATAAGCTAAACGAATTAAAATAATCATTTCCATGAGTTCGAATTATAGAAACCAAAATTGATAATCCTAATACACCTTCACAAACTGCAAATGAAAGAAAAATTATAGTCATATATAACTCTCCTCTTATTATTAAATAAAAATACAAAATAATGAACAAAACTAAAACAATAAATTCTAAACTTAATAAAGTTACTAATAAATGTTTACGTCCAGAAGAAAAAACTCACAAACCACATAAAAATATAAAACAGAATATTATTAACATTTTTTAGTTTTAATAGTTTACTAAAAACACTGGTCTTGTAAACCAAAATTAAGAATAATTACTTTTAAAACTTCAAAGGAAAGAAACTCATCATTAATTCCCAAAATTAATATTTTATTATTAAACTACCCTTTGATATTTTATACTCATTATTAACTATAAGTTTAATTTTAAGAATTACTTTGATATTTTTAAATCATCCAATATCTATAGGTTTTATCCTTTTTATTCAAACGTTATGCTTATGTTTTATTAGAGGTTTTATATCACTAAGATTTTGATTCTCCTATGTATTATTATTAATTTATTTAGGAGGAATATTAATTTTATTTATATACATTACCAGATTAGCCTCAAACGAATTATTTTCTTATTCAAATAAAATCTTATTAATTATTTTTTTAGTACCCCTTATCTTAAGACTTATCCATTATACAAGTTTTTATCATCAAACAAACTTATATGAAAATATAGAAAATAGAATAAATTTCATCTTTATACCAAACAACTTTTTATTAAAGATATATAATTATCCTAATAACATTATTACAATCTTAATTGCCTGTTATTTATTTTTAGCTTTAATTGCAGTTGTTAAGATAACAAATATTTTCAAAGGACCCCTACGACAAATAAATTAATAATGTTTAAACCCTTACGAAAAACTCACCCTATCATCAAAATTTCCAATAATGCCTTAGCAGATTTACCTTCACCATCAAATATTAGATCATGATGAAATTTTGGTTCCCTTCTAGGCTTATGTTTAATTCTTCAAGTAATTACAGGACTATTTTTAGCTATACACTATTCAGCTCATATTGATTTAGCCTTCTCAAGAGTGATTCACATTTGTCGAGATGTAAATTATGGGTGACTTTTGCGAATTCTTCATGCTAATGGTGCTTCAATATTTTTTATTTGTATTTATCTCCATATTGGACGGGGGATATATTACGGATCATATAAATTTTATTATACATGAATAATTGGAGTAATAATTCTATTTTTATTAATAGCGACAGCATTTATAGGTTATGTATTACCATGAGGTCAAATATCATTTTGAGGTGCCACAGTAATTACTAATTTATTATCTACTATTCCCGGATTTGGAAATGAATTAGTTCAATGAGTTTGAGGGGGTTTTGCAGTAGATAATGCTACCCTAAACCGATTTTTCACATTTCATTTTATTTTACCATTTATTATTGTAGCAATAATTGCAACCCACTTATTATTTCTTCACCAAACTGGATCAAATAATCCCTTAGGTACTGATAGTAATATTGATAAAATTCCGTTTCACCCTTATTTCACATTTAAAGATATCTTAGGATTCATTATATTATTAACACTTCTATCCCTCTTATCTCTTAAAGAACCCTATATCCTAGGAGATCCTGATAATTTTATTCCTGCTAACCCTCTTGTTACTCCAGTCCACATTCAACCAGAATGATACTTTTTATTTGCTTATGCTATTTTACGCTCTATCCCTAACAAGCTAGGAGGGGTGATTGCTCTAGTAATATCTATTATAATTCTAATTATTATACCTTTCATAGATACTAACTTACGAAGATTTCAGTATCATCCTATTAATCAATTCATATTTTGATTTATAATTATAACTATTACACTTTTAACATGAATTGGAGCACGCCCTGTTGAAGACCCTTTTATCCTAGTAGGACAAATTCTCACCATTATATACTTCCTTTTCTATATCATAAACATTTTAATTATCAAGGTATGAGACCATATTATACAATAATAAATAAGTTATAAACTTAATGTTAAGATTATGTCTTGAAATCATAATAAAGGGGTTAAATTCCCTTATTAACTTTACTTCATCCAACCCTTAAAATAAAATTTTTAACCCTAAATAAAAAAACAAAAAATTTAAAGATAAAGGTAAAAAGCTTTTTCACGCTAAATATATTAATTTATCATATCGATATCGAGGAAGAGTTCCACGAACTCAAATATATATAAAAGCAATAAAAATCAACTTCAAATAAAATACAAATGAATCCAAATTAGAACCCAAAAAAATTACACATATTAACATTCTTATAAATAAAATTCTTGAATATTCTCCTAAGAATATTAATGCAAAGCCTCCCCCTCCATACTCAATATTAAAACCTGAAACTAATTCTGACTCTCCTTCAGCAAAATCAAAAGGTCTACGATTTGTTTCTGCTAAACATGACACTATTCACACATTACACAATGGTAAGTATAAAAATAAAAATCAAATATTCATCTGATAATAAAAAAAATCCATTAATGAATAACTTCCAATTAAAAAAACAAAAGATAATAAAATTAAGGCTAATCTTACCTCATAAGAAATTGTCTGAGCAATTGCTCGTAAACTCCCTAATAAAGCATAATTAGAATTTGAAGATCATCCAGCTAACATAACTGCATAAACACCTAAACTTGTACATACTAAAAAAAACAACAAACCCAAATTAAAAGTAAATAAACCTCTTATATAAGGAAATAATATTCATAAGATTAAAGATAAAAATAAAGTAAATATTGGACATACATAATATAATGAATAATTCGAAACTAAAGGATTTATATATTCCCTACAAAACAACTTTACAGCATCACTGAATGGTTGTAAAATACCAATAAATCCAATCTTATTAGGACCCTTACGAATATGAATGTATCCTAAAACCCTACGTTCTAATAAAGTAAAAAAAGCTACCCCAACTATAACAAAAATAACTAAAATTAACCCTACTAATATTATTAAAAGTACTTCACTAATTAACATTTACTATCTATAATTCAATTTATCTATTTAGGTTCTAAACCTATTACACTTTACTCTGCCAAAATAGTTTTATTAATAATATTCTTTTTCCTTCATAAAAATTATTTCAAATATTTGGTCCTTTCGTACTAAAATATTCTAATTAATTAAAGATAGAAACCAACCTGGCTCACGCCGGTTTAAACTCAGATCATGTAAGAATTTTAAGGTCGAACAGACCTAGTTATTAAACATCTACATTCAAAACTAATCTTAATCCAACATCGAGGTCATAATCTTTTTTTTCAATATGAACTTTCAAAAAAAATTATGCTGTTATCCCTAAGGTAATTTAATCTTCAAATCATTAACCATGGATCTTAAAACTTAATATAATATTAAACTCAAAAAAGAGTTCATTTTATCTTTTAATCACCCCAATTAAATATATTTCACATAGATAATAAATTCATTATTTTAACTTTATTAAATTCAAATATATATTAAACTCTATAGGGTCTTCTCGTCCCTTAATTTTATTTAAGTATTTTTACTTAAAATCTAAGTTAAATTAAATTATTATAAAATAGTCTTTACTTCATCCAACCCTTCATACTAGCCCTCAATTAAAAGACTAATGATTACGCTACCTTTGCACAGTCAATATACTGCGGCCCTTTAAACCACATTCAGTGGGCAGGTTAAATCTCTTATTTCCCCAAGAAACCATGTTTTTAATAAACAGGTGAGTAAATTATTTGCCTAATTCCTCATATTAACATTACATTTCATTCACAAAATAATCCAGACTTTACTAATTAAATCATACTTACTCACTTTTTTTTATTTAACTTAACATTTTAATATACATACTAAATAATAACTAAAAATTTTAAACAAAATAAGAATTAAATTTCAACATCCTTAAATTAATAACAAATTCTAAATTTATACAATTCTTCATTAATAAATTATTATAAATATTCTTCAAAAAGATTTTCCCTTTTAAAGTCAAGAAAACAAAATATTTAAATATTTAAATATTTAATAAATTATATTCTCTAAATTAAATCTATTTATTAAAAAACTAGATATTATTGAAAACGATTAACATTTCACTACTAATTAATAATTCTTAATATTTATAATACAATAATTAAAACTATTAATTAAATCTTTCAAATTCAAGAAAAAAAAAATTAATTATTAATTTCGATATACTCTGAAACACAAGATACAATAAAAAAAATCAACTTAAATTAATTATATAATATTAAACTTCTTTTACAATACTAATTTACTATAGTATTAATTATTATCTCAATTGATAATACAATAACAAAATTTTAATTAATTTATATTCTAAATAATATCAAATCAAAAACAATCTTATGAATTTCAAATTATATTGAATTGCACAATAATAATTTTCAATGTAAATGAAAATCCTTACCTTTAAGTTTAATTTGTTCAAGTGACTTTCTAGGCACACCTTCCGGTACGCCTACTTTGTTACGACTTATCTCATCTTAATATGAGAGTGACGGGCGATGTGTACATATTATAGAGCTATTAATCATTTTAACAAACTTCATAAAATTACAATTAAATCCACCTTTAAATTACTTTCCAATAATTATCCATAATAAATACAATTTATTGTAGTCCATCATCAAACTTACTTATTACTGCACCTTGACTTGAAATATTATATATTTTTATGCTCAGAAAATTATCTAAAAAAATATTCTCAAACAGCGGTATACAAAAATATAAAGTAAGTAAGGTCCAACGCGGATTATCAATTAAATAATAGACTCCTCTAAATAGGTTATAATACCGTCAAATTCTTTAAGTTTCAAGATCATATCTGCTAATACTTTAGTTTATTAATTTACCATAAAAATAATAGGGTATCTAATCCTAGTTTACCTTTTTAATCTCATAATCCATCATACTAACTAATAATTAATTAAAATATTTTAAAAACATTTCACCTTAAAATACTTTAATCATCATTTAACATTAATTAACATTAATTATTTAAAAACCAATATTATTTATTTAATTTCACTGTATCACCGCGGATGCTGGCACAATTTTTACCAAACTTTTATAACATTCACTAAATCTAAAATAATTTAATATCAATAAAATTACCTACTGCACAATTAACTTTATAAACCTCATTGAGAAAACTATAATATACACACATTCACATTCAGAATAACATTAAAACAAATAATCAAAGCAAGAATAAAACTAAATATTTTTTCCAAAAATCGAAGACCTGACATTGGAACACCTAACTTTTTTAATGATTTTTGTATACCTACAAATATTTGATTAAATTAATCATTGCTTCAATATATAAACTTTCAAAAAAAAAATCCACTTTACCCAAAAAATAAGTATGCTTTTTTTAAAACGTTTTCCTCCCTAGAAGTAGGAATTAAGTGCCCAATTCAATTCGATTTAATGTTTTTTTTTAAACATTGGTCTACTAAAGTATATTTAGTATATAATAAATTTAATTATATATATGTCTTATCAAAGATTAATAAAAAAAATATTCTCAAACAGCGGTATACAAAAATATAAAGTAAGTAAGGTCCAACGCGGATTATCAATTAAATAATAGACTCCTCTAAATA

4. I recommend one more round of editing for writing clarity. For example, Lines 50-51 are a little confusing, and should probably be split up into two sentences.

Reply: Thanks for your suggestion. Lines 50-51 have been split up into two sentences. Please check lines 56-57.

---

## [Editor Report · Decision Letter 2]

7 Jul 2021

Complete mitochondrial genomes of four species of praying mantises (Dictyoptera, Mantidae) with ribosomal second structure, evolutionary and phylogenetic analyses

PONE-D-21-03889R2

Dear Dr. Smagghe,

We’re pleased to inform you that your manuscript has been judged scientifically suitable for publication and will be formally accepted for publication once it meets all outstanding technical requirements.

Kind regards,

Ben J Mans, PhD

Academic Editor

PLOS ONE
---

## [Editor Report · Acceptance letter]

21 Oct 2021

PONE-D-21-03889R2 

Complete mitochondrial genomes of four species of praying mantises (Dictyoptera, Mantidae) with ribosomal second structure, evolutionary and phylogenetic analyses 

Dear Dr. Smagghe:

I'm pleased to inform you that your manuscript has been deemed suitable for publication in PLOS ONE. Congratulations! Your manuscript is now with our production department. 

Kind regards, 

on behalf of

Dr. Ben J Mans 

Academic Editor

PLOS ONE